# Social groups buffer maternal loss in mountain gorillas

**Robin E Morrison[1,2]\*, Winnie Eckardt[1], Fernando Colchero[3,4], Veronica Vecellio[1], Tara S Stoinski[1]**

[1]Dian Fossey Gorilla Fund, Musanze, Rwanda; [2]Centre for Research in Animal Behaviour, University of Exeter, Exeter, United Kingdom; [3]Department of Mathematics and Computer Science, University of Southern Denmark, Odense, Denmark; [4]Interdisciplinary Center on Population Dynamics, University of Southern Denmark, Odense, Denmark

**Abstract** Mothers are crucial for mammals' survival before nutritional independence, but many social mammals reside with their mothers long after. In these species the social adversity caused by maternal loss later in life can dramatically reduce fitness. However, in some human populations these negative consequences can be overcome by care from other group members. We investigated the consequences of maternal loss in mountain gorillas and found no discernible fitness costs to maternal loss through survival, age at first birth, or survival of first offspring through infancy. Social network analysis revealed that relationships with other group members, particularly dominant males and those close in age, strengthened following maternal loss. In contrast to most social mammals, where maternal loss causes considerable social adversity, in mountain gorillas, as in certain human populations, this may be buffered by relationships within cohesive social groups, breaking the link between maternal loss, increased social adversity, and decreased fitness.

**\*For correspondence:** rmorrison@gorillafund.org

## Introduction

Maternal loss, along with a number of other indicators of early-life adversity, is one of the strongest predictors of lifespan in humans and other social mammals (*Snyder-Mackler et al., 2020*). In mammals, mothers are vital for the survival of young offspring, providing nutrition, thermoregulation, and protection (*Clutton-Brock, 1991*). In some species, particularly social species with slow life histories (*Mitani et al., 2013*), mothers continue to provide benefits to their co-residing offspring throughout immaturity and even into adulthood (*Surbeck et al., 2019*; *Surbeck et al., 2011*; *Andres et al., 2013*; *Stanton et al., 2020*). The active support of mothers can increase the rank of their offspring (*Strauss et al., 2020*; *East et al., 2009*; *Maestripieri and Mateo, 2009*; *Lea et al., 2014*) and improve their integration in the group (*Tung et al., 2016*), both of which are linked with greater survival (*Snyder-Mackler et al., 2020*; *Archie et al., 2014*). Maternal presence can also influence nutrition at these later stages of development by buffering against feeding competition (*Samuni et al., 2020*), providing access to valuable ecological knowledge (*Stanton et al., 2020*; *Foley et al., 2008*; *Brent et al., 2015*) or increasing opportunities for the social learning of complex feeding techniques (*Lonsdorf et al., 2004*; *Estienne et al., 2019*).

In social mammals maternal loss can therefore reduce the fitness of offspring across a broad age range through long-term effects on their social environment, negatively influencing their social integration and social status throughout their lives (*Snyder-Mackler et al., 2020*; *Strauss et al., 2020*; *Tung et al., 2016*). Multiple studies have now confirmed effects on survival for individuals orphaned well past the period of nutritional dependency, with these negative changes to their social environment, often termed social adversity, posited to be the key mechanism by which this occurs

**eLife digest** Most mammals depend entirely upon their mothers when they are born. In these species, losing a mother at a young age has dramatic consequences for survival. In cases where orphaned individuals do reach adulthood, they often suffer negative effects, like reduced reproductive success or lower social status. But this is not the case for humans. If a child loses their mother, relatives, friends and the wider community can take over. This does not tend to happen in nature. Even our closest relatives, chimpanzees, are much less likely to survive if their mothers die before they reach adolescence.

Although orphan survival is not the norm for mammals, humans may not be entirely unique. Mountain gorillas also live in stable family groups, usually with a dominant male and one or more females who care for their offspring for between 8 and 15 years. It is possible that gorillas may also be able to provide community support to orphans, which could buffer the costs of losing a mother, just as it does in humans.

To answer this question, Morrison et al. examined 53 years of data collected by the Dian Fossey Gorilla Fund to assess the effects of maternal loss in mountain gorillas. The analysis examined survival, reproduction and changes in social relationships. This revealed that, like humans, young gorillas that lose their mothers are not at a greater risk of dying. There is also no clear long-term effect on their ability to reproduce. In fact, gorillas who lost their mothers ended up with stronger social relationships, especially with the dominant male of the group and young gorillas around the same age. It seems that gorilla social groups, like human families, provide support to young group members that lose their mothers.

These findings suggest that the human ability to care for others in times of need may not be unique. It is possible that the tendency to care for orphaned young has its origins in our evolutionary past. Understanding this in more depth could provide clues into the social mechanisms that help to overcome early life adversity, and have a positive impact on future health and survival.

(*Andres et al., 2013*; *Stanton et al., 2020*; *Tung et al., 2016*; *Watts et al., 2009*; *Foster et al., 2012*). In social mammals, the social environment can have extreme consequences for health, fitness, and lifespan, mediated through pathways such as chronic stress, immune function, or environmental exposure (*Snyder-Mackler et al., 2020*).

The impact of maternal loss varies based on the loss in benefits relative to individuals with mothers present. In species with sex-biased dispersal, the consequences of maternal loss can differ between the sexes due to longer periods for potential investment in non-dispersing offspring (*Clutton-Brock, 1991*; *Fairbanks, 2009*; *Altmann and Alberts, 2005*; *Greenwood, 1980*). In female-philopatric red deer (*Cervus elaphus*), maternal loss increases mortality for males and females, but this effect is only detectable in males under 2 years of age, whilst for females, maternal benefits continue throughout their lives (*Andres et al., 2013*). In male-philopatric chimpanzees (*Pan troglodytes*), males that suffer maternal loss before reaching 15 years of age have lower survival, whilst females only show reduced survival from maternal loss under the age of 10 (*Nakamura et al., 2014*; *Stanton et al., 2020*). In killer whales (*Orcinus orca*), where neither sex disperses, maternal loss when offspring are over 30 years old reduces survival for both sexes, although considerably more so for males (*Foster et al., 2012*). However, maternal loss between 15 and 30 years appears to reduce male but not female survival. This is thought to be due to higher maternal investment in males which mate outside the group and whose offspring therefore do not increase within-group feeding competition.

As a result of the numerous benefits mothers can provide, it is not surprising that maternal loss not only influences offspring survival but can also impact other components of their offspring's fitness, such as reproduction and the survival of grand-offspring. Male bonobos (*Pan paniscus*) residing in groups with their mothers sire three times the number of offspring (*Surbeck et al., 2019*), whilst maternal loss before weaning negatively affects antler development in male red deer – a trait found to correlate with reproductive success (*Andres et al., 2013*). In chimpanzees, females mature faster, first give birth younger (*Walker et al., 2018*) and enter the dominance hierarchy higher (*Foerster et al., 2016*) if their mothers are present which is expected to considerably increase their

lifetime reproductive success. In savannah baboons (*Papio cynocephalus*), if mothers had themselves suffered maternal loss in the first 4 years of their life, their offspring had 48% higher mortality throughout the first 4 years of their life, suggesting an intergenerational effect of maternal loss driven by lifelong developmental constraints (*Zipple et al., 2019*). Maternal loss in social mammals with extended maternal care can therefore have long-term fitness consequences mediated through multiple pathways that detrimentally affect survival and reproduction.

Due to the extended periods of mother–offspring co-residence and the important social support mothers can provide, highly social species often have the most to lose from maternal loss. However, social groups also provide the potential for support from other group members following maternal loss. Both kin and non-kin group members have been suggested to compensate for the loss of close kin to varying extents (*Hamilton et al., 1982*; *Engh et al., 2006*; *Goldenberg and Wittemyer, 2017*; *Reddy and Mitani, 2019*) and the strengthening of relationships with remaining group members may buffer against changing social environments (*Firth et al., 2017*). In chacma baboons (*Papio ursinus*), social support from group members is thought to alleviate the stress of losing a close relative (*Engh et al., 2006*). Similar social support has been suggested in African savannah elephants (*Loxodonta africana*) which associate more with group members of a similar age and siblings in response to maternal loss (*Goldenberg and Wittemyer, 2017*). However, these orphaned elephants interact less with matriarchs which may decrease their access to key knowledge and high-quality resource patches. In chimpanzees, older siblings can 'adopt' younger siblings after maternal loss, increasing their social contact and showing heightened vigilance in dangerous situations (*Reddy and Mitani, 2019*; *Hobaiter et al., 2014*). But despite these compensatory social behaviours, the negative consequences of maternal loss post-weaning are well documented in all three genera (*Stanton et al., 2020*; *Tung et al., 2016*; *Samuni et al., 2020*; *Nakamura et al., 2014*; *Goldenberg and Wittemyer, 2017*; *Goldenberg and Wittemyer, 2018*).

Humans (*Homo sapiens*) are a rare example of a group-living mammal in which compensatory social behaviours have been suggested to have the capacity to consistently overcome these negative consequences of maternal loss. In a meta-analysis of historic and contemporary human populations, the death of a mother was associated with increased child mortality in all 28 populations studied (*Sear and Mace, 2008*). However, this effect appeared to decline substantially with age, disappearing for children that suffered maternal loss over 2 years of age in 5 of the 11 populations in which it was investigated (*Sear and Mace, 2008*). This reduced mortality was thought to be due to the care provided by other kin, particularly after weaning, suggesting that social buffering from other group members can overcome the negative effect of maternal loss on survival in certain circumstances. Whilst the effects of care from specific kin members varied across populations, at least one kin member significantly impacted child survival in all studies (*Sear and Mace, 2008*), and there is evidence for the importance of maternal grandmothers (*Lahdenperä et al., 2004*; *Sear et al., 2000*) and fathers (*Hurtado and Hill, 1992*; *Hill and Hurtado, 2017*) in particular. In killer whales, maternal grandmothers, especially those that are post-reproductive, are also known to improve grand-offspring survival (*Nattrass et al., 2019*). Whilst the specific effect of killer whale grandmothers on orphan survival has not been investigated, this finding suggests that killer whales may represent a further species in which care from other kin has the capacity to overcome the effects of maternal loss. In humans, there is also evidence for the benefits of care provided by non-kin such as stepmothers (*Andersson et al., 1996*; *Campbell and Lee, 2002*) and through the modern practices of non-kin adoption (*Bentley and Mace, 2009*).

Mountain gorillas (*Gorilla beringei beringei*) show extended maternal care with offspring remaining in their natal groups at least until sexual maturity and approximately half remaining beyond sexual maturity (48% of females [*Robbins et al., 2009a*] and 55% of males [*Stoinski et al., 2009a*]). Females that disperse from their natal group tend to do so earlier (mean age of 7.9 years [*Robbins et al., 2009a*]) than males (mean age of 15.3 years [*Stoinski et al., 2009a*]) and therefore have a shorter period of potential maternal investment. The complexity of gorilla social structure with numerous types of differentiated social relationship both within and among groups (*Morrison et al., 2019*; *Mirville et al., 2018*; *Morrison et al., 2020a*; *Morrison et al., 2020b*) suggests that detrimental long-term effects on individual gorillas' social environments could have particularly negative fitness consequences. However, these stable, cohesive, social groups also have the potential to provide a social buffer to the negative consequences of maternal loss. Mountain gorilla groups either contain a single adult male (approximately 64% of groups) or multiple adult males

(approximately 36% of groups) (*Gray, 2010*), at least one adult female, and their offspring (*Robbins, 1995*). Single male groups are polygynous whilst multimale groups have high reproductive skew towards the dominant male who sires the majority of offspring (*Nsubuga et al., 2008*; *Stoinski, 2009b*; *Bradley et al., 2005*). Infants (<4 years of age) are nutritionally dependent on their mothers until being weaned at a mean of 3.3 years (*Eckardt et al., 2016*) and are reliant on their mothers for thermoregulation and transport, being carried for prolonged periods (*Breuer et al., 2009*). Juveniles (4–6 years old) are nutritionally independent but remain in close proximity to their mothers the majority of the time (*Breuer et al., 2009*). Dominant males' primary form of care is through protection from out-group males and potential predators (*Harcourt and Greenberg, 2001*). But they also show high levels of affiliative behaviour towards infants, grooming and resting in contact with them, with no evidence that they discriminate between infants based on paternity (*Rosenbaum et al., 2018*; *Rosenbaum et al., 2015*).

In this study, we use the long-term demographic records of the Dian Fossey Gorilla Fund's Karisoke Research Center collected over 53 years (1967–2019) to (A) quantify the effects of maternal loss on multiple fitness measures: survival, female age at first birth, female survival of first offspring through infancy, and male dominance; and (B) investigate the social responses of group members to maternal loss by immature gorillas. We hypothesize that as demonstrated in chimpanzees (*Stanton et al., 2020*), gorillas may face greater fitness costs if they suffer maternal loss at an earlier age and that males may face greater costs from maternal loss than females due to their longer periods of mother–offspring co-residence. Alternatively, as observed in many human populations, the cohesive, stable social groups of mountain gorillas may enable social buffering from group members to compensate for the social costs of maternal loss with minimal fitness consequences to maternal loss. In particular, dominant males may take on crucial roles in buffering the social adversity faced by maternal orphans (hereafter, orphans), as past research has demonstrated the strong bond between dominant males and young orphans who may regularly share a nest at night (*Robbins et al., 2005*; *Gatesire et al., 2016*).

## Results

### Effect of maternal loss on survival

To determine the effect of maternal loss on survival, we carried out a Cox-proportional hazards analysis separating individuals into four orphan classes based on their age when their mother died: (a) infants (2–4 years old), (b) juveniles (4–6 years old), (c) subadults (6–8 years old), and (d) non-orphans (>8 years old) if their mothers died after they had reached maturity. Due to the small sample sizes for the juvenile and subadult classes, analysis was also run with these two classes merged. We found no significant differences in survival between all orphan classes and the non-orphan class for both sexes irrespective of using three or four classes (*Table 1*, *Figure 1*, *Figure 1—figure supplement 1*, *Supplementary file 1* -Table 1). Bayesian survival trajectory analysis (*Colchero and Clark, 2012*; *Colchero et al., 2012*) showed similar results, whereby the model with highest support for both sexes was the null model without orphan classes as covariates (*Table 2*).

**Table 1.** Cox-proportional hazards models showing the effects of the four maternal loss categories: infants, juveniles, subadults, and non-orphans, for each sex (98 females and 102 males), on survival. All results are relative to the non-orphan class.

| Age-class | Females | | Males | |
|---|---|---|---|---|
| | Est ± SE | p | Est ± SE | p |
| Infants | 0.73 ± 0.592 | 0.218 | −0.34 ± 0.551 | 0.540 |
| Juveniles | 1.22 ± 1.170 | 0.298 | 0.05 ± 1.060 | 0.959 |
| Subadults | 1.77 ± 1.240 | 0.152 | 0.59 ± 1.110 | 0.591 |

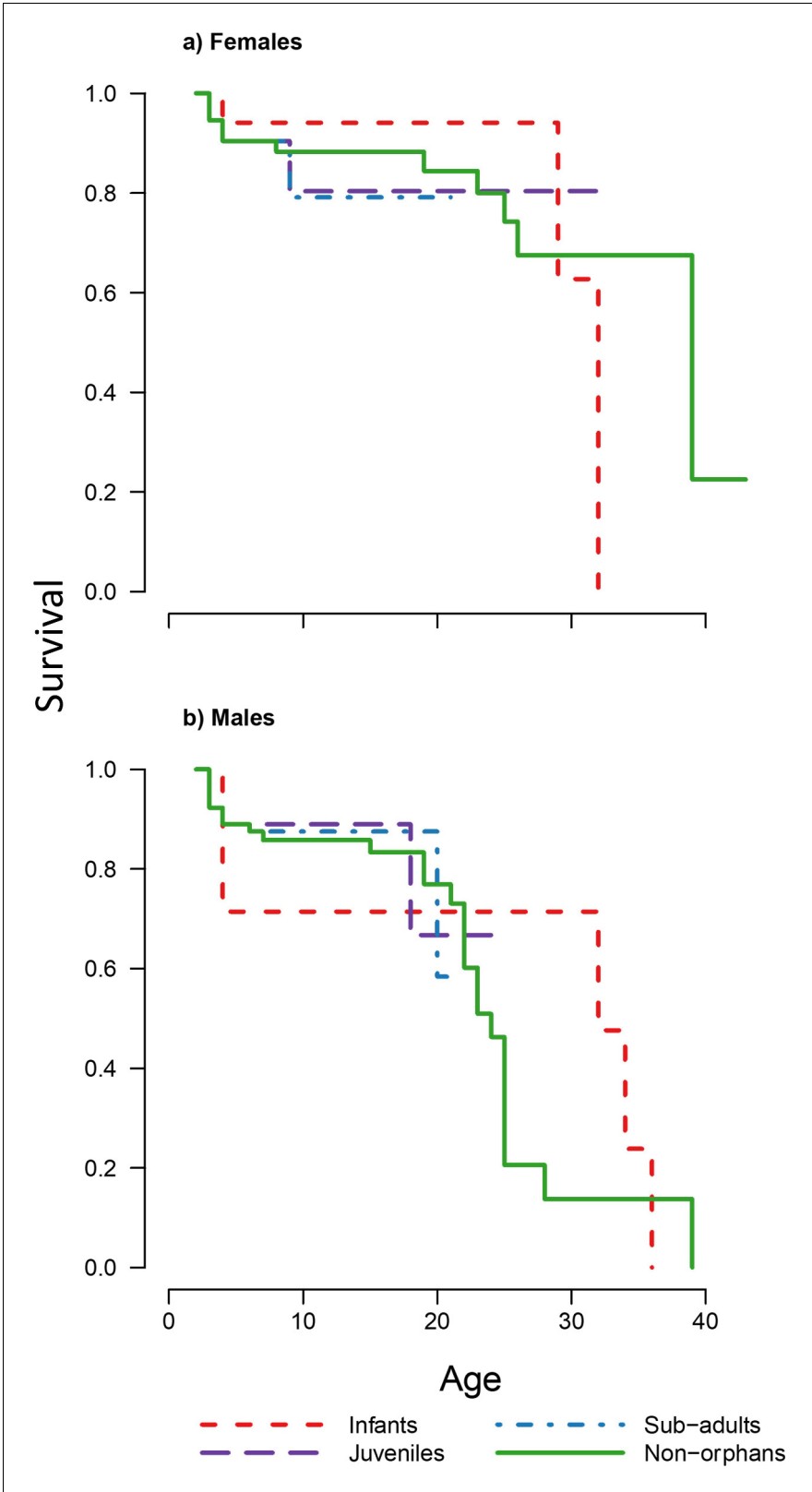

**Figure 1.** Survivorship curves for each of the four maternal loss categories for each sex. Plots show the proportion of surviving (a) females (n = 98) and (b) males (n = 102) that suffered maternal loss as infants, juveniles, and subadults compared to non-orphans that did not suffer maternal loss under the age of 8 years. *Figure 1—figure supplement 1* shows survivorship curves of each orphan age and sex class plotted separately against non-orphans *Figure 1 continued on next page*

*Figure 1 continued*

for further clarity. Grey dashed line indicates the age from which orphan and non-orphan survival are modelled separately.

The online version of this article includes the following figure supplement(s) for figure 1:

**Figure supplement 1.** Orphan survivorship curves separated by age and sex class.

## Effect of maternal loss on dispersal

We examined the effect of maternal loss on the likelihood of females dispersing before giving birth to their first offspring using a binomial generalized linear model (n = 51). Female orphans were not significantly more likely to disperse from their natal group prior to first birth than female non-orphans (*Table 3*). However, there was a close to significant increase in the likelihood of dispersal for females that lost their mothers as juveniles or subadults. 37.5% of non-orphan females (n = 32) dispersed prior to their first birth (mean dispersal age ± SD: 7.96 ± 1.55 years) compared to 54.5% (n = 11) of infant orphans (dispersal age: 7.75 ± 0.60 years) and 75.0% (n = 8) of juvenile and sub-adult orphans (dispersal age: 8.21 ± 2.36 years). We examined the effect of maternal loss on male dispersal based on whether a male had dispersed from their natal group prior to the age of 16 years (the median age of male dispersal). Only three males that reached the age of 16 had lost their mothers as infants, but all three remained in their natal group. Due to this small sample size this was not examined statistically. However, a binomial generalized linear model demonstrated that juvenile and subadult orphan males were significantly more likely to disperse before reaching 16 years of age (84.6%, n = 13) than non-orphan males (37.5%, n = 40, *Table 3*).

## Effect of maternal loss on female reproduction

Using a generalized linear model, we found that maternal loss had no significant effect on the age at which females first gave birth (*Supplementary file 1* - Table 2, n = 53). The mean age at first birth (± SD) for non-orphans was 10.24 ± 1.61 years compared to 9.72 ± 0.73 years for those orphaned as infants and 9.67 ± 1.82 years for those orphaned as juveniles or subadults. After accounting for age at first birth and dispersal, there was also no evidence that maternal loss influenced whether a female's first offspring survived infancy (*Supplementary file 1* - Table 3, n = 50, binomial generalized linear model). 51.5% of non-orphan females' first-born offspring (n = 33) survived infancy compared to 60% (n = 10) of first-born offspring of those orphaned as infants and 57.1% (n = 7) of first-born offspring of those orphaned as juveniles or subadults.

## Effect of maternal loss on male dominance attainment

The mean age (± SD) at which a male first became the dominant male of a group was 17.88 (± 2.56) years. The oldest that a male first reached dominance was 22.99 years, with all males that had not become dominant by this age, never reaching dominance. We therefore compared the proportion of males over the age of 23 that had attained dominance in each orphan class. 52% of non-orphan males had become the dominant male of a group for at least 6 months (n = 21) by the age of 23

**Table 2.** Deviance information criterion (DIC) for the three models tested.

(a) No covariates (i.e. null model where all individuals have the same hazard rate); (b) proportional hazards (where mortality differs proportionally between orphan classes); (c) covariates modifying all Siler mortality parameters (where each orphan class has a different age-specific mortality), for each sex (98 females and 102 males). The delta DIC shows the difference in DIC from the model with lowest DIC.

| Model | Females | | Males | |
|---|---|---|---|---|
| | DIC | Δ DIC | DIC | Δ DIC |
| No covariates | 371.46 | 0 | 486.08 | 0 |
| Prop. hazards | 374.16 | 2.71 | 486.44 | 0.36 |
| All mortality parameters | 377.04 | 5.59 | 499.46 | 13.38 |

**Table 3.** The influence of age at maternal loss (infant or juvenile/subadult (J/SA)) relative to non-orphans on a female's decision to disperse from their natal group prior to their first birth and a male's decision to disperse prior to the age of 16, modelled using binomial generalized linear models.

|  | Females (n = 51) | | | Males (n = 53) | | |
|---|---|---|---|---|---|---|
|  | Est ± SE | T | P | Est ± SE | T | p |
| Intercept | −0.511 ± 0.365 | −1.399 | 0.162 | −0.511 ± 0.327 | −1.564 | 0.118 |
| Infant | 0.693 ± 0.707 | 0.980 | 0.327 | - | - | - |
| J/SA | 1.609 ± 0.894 | 1.799 | 0.072 | 2.216 ± 0.835 | 2.653 | 0.008 |

compared to all three infant-orphaned males and no juvenile- or subadult-orphaned males (n = 5). This finding suggested maternal loss as a juvenile or subadult male could limit a gorilla's ability to become dominant. However, this could also be purely an artefact of the small sample sizes. We therefore examined the dominance status of the seven juvenile- or subadult-orphaned males that were over the age of 16 but had not yet reached 23 years by the end of the study period. 71% (n = 7) of these males had already become dominant despite their younger age, indicating the capacity for males orphaned in any age category to become dominant later in life.

## Changes in network position following maternal loss

The social responses of group members to 21 incidents of maternal loss were investigated (*Supplementary file 1* - Table 4). Focal data collected daily in each group were used to construct social networks based on (a) 2 m proximity and (b) affiliative contact (resting, playing, or feeding in physical contact and grooming, but excluding physical aggression) for the 6 months leading up to a maternal loss incident and the 6 months immediately after a maternal loss incident. In the 6 months prior to maternal loss, orphans had spent a mean of 13% (± 8, n = 31) of their time while monitored in affiliative contact with their mother and 39% (± 16, n = 31) of their time within proximity (<2 m) of their mother, who was predominantly their closest social partner (*Table 4*). After maternal loss, orphan's affiliative contact with other group members increased on average by 28% and the proportion of their time spent within 2 m of other group members increased on average by 42%. Overall, this resulted in a net decrease in orphan's affiliative contact of 31% and a net increase in orphan's proximity to others of 11% following maternal loss.

To investigate how these changes influenced the social network position of orphans we compared orphan's change in binary degree (number of connections), weighted degree (strength of connections), and eigenvector centrality (how connected they were to other well-connected individuals) with that of non-orphans within the same networks (n = 136). These networks included only individuals that were present both before and after the maternal loss incident (excluding the mothers of orphans) to enable direct comparison. Using generalized additive mixed models (GAMMs) with node-level permutations of orphan status we found that within proximity-based networks, orphans' eigenvector centrality (Est = 0.169 ± 0.037, t = 4.594, p < 0.001, $P_{null}$ < 0.001) and weighted degree (Est = 0.075 ± 0.032, t = 2.337, p=0.021, $P_{null}$ = 0.024) increased significantly more after maternal loss than non-orphans within the same pair of networks. This led to orphans and non-orphans having similar weighted degree and centrality values following maternal loss despite orphans losing a key social partner (*Figure 2*, *Supplementary file 1* - Table 5). However, within contact-based networks orphans did not show these same gains, with no significant change in eigenvector centrality (Est

**Table 4.** The percentage of gorilla orphans for which their mother was their closest social partner prior to maternal loss based on affiliative contact and proximity within 2 m.

|  | Contact | Proximity |
|---|---|---|
| Infants (n = 9) | 89% | 78% |
| Juveniles (n = 14) | 86% | 79% |
| Subadults (n = 8) | 25% | 75% |

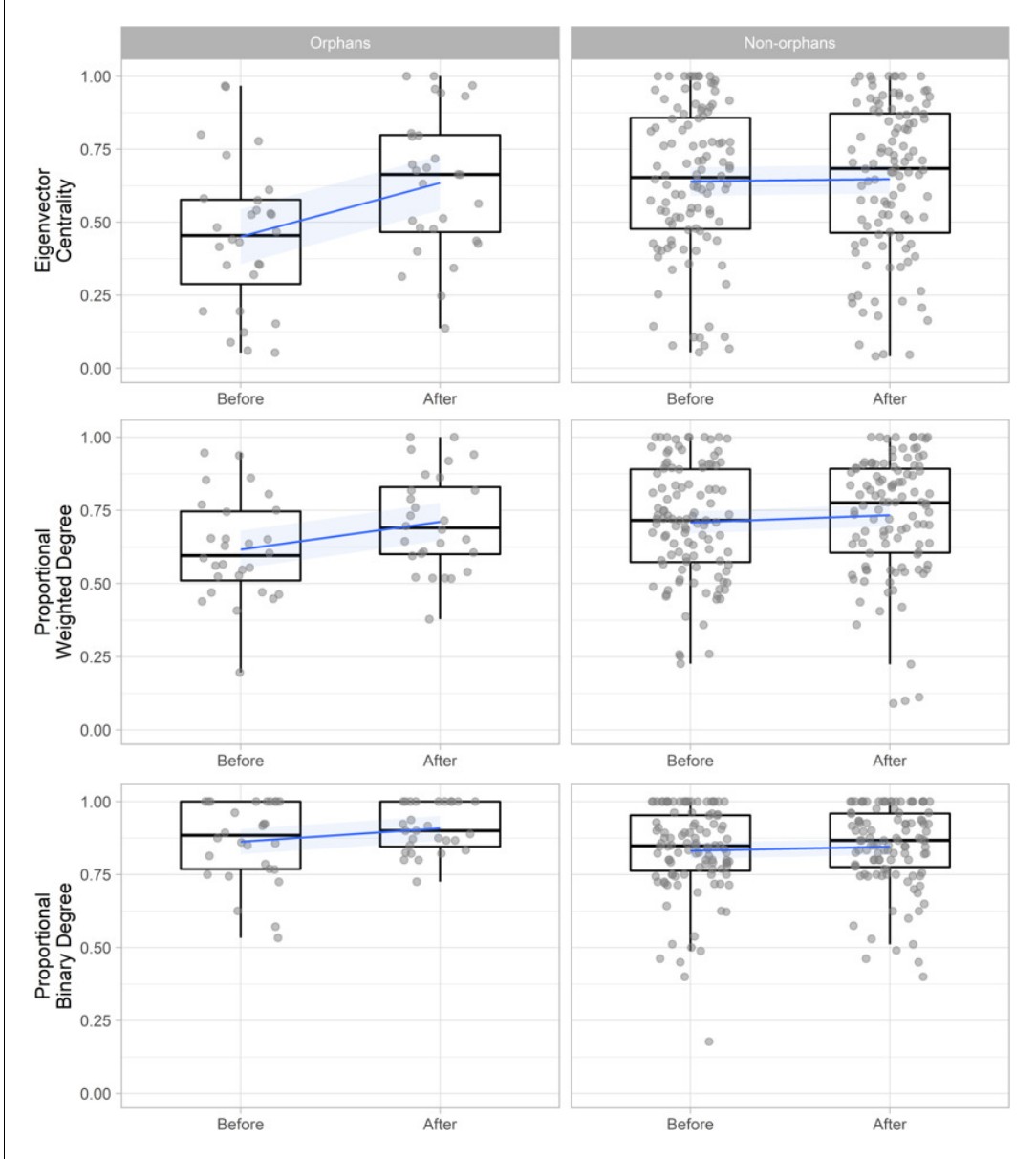

**Figure 2.** The change in proximity-based social network position (eigenvector centrality, weighted degree, and binary degree) for orphaned (n = 28) and non-orphaned immature gorillas (n = 108) within the same group in the 6 months before an incident of maternal loss and the 6 months after. Weighted degree and binary degree values are calculated as proportions of the greatest value observed within their specific network to enable comparison across multiple networks (multiple incidents of maternal loss). Networks include only individuals that were present both before and after maternal loss and therefore exclude the mothers of orphans. Blue lines and shading indicate mean and 95% confidence interval. *Figure 2—figure supplement 1* shows the change in affiliative contact-based social network position. Source data available in file: '*Figure 2—source data 1*'. The online version of this article includes the following source data and figure supplement(s) for figure 2:

**Source data 1.** Changes in the network position of orphaned and non-orphaned immature gorillas following an incident of maternal loss.

**Figure supplement 1.** The change in affiliative contact-based social network position (eigenvector centrality, proportional weighted degree, and proportional binary degree) for orphaned and non-orphaned immature gorillas within the same group in the 6 months before an incident of maternal loss and the 6 months after. Blue lines and shading indicate mean and 95% confidence interval.

= $-0.034 \pm 0.057$, t = $-0.593$, p = 0.554, $P_{null}$ = 0.598) or weighted degree (Est = $0.031 \pm 0.047$, t = 0.656, p = 0.513, $P_{null}$ = 0.442) relative to non-orphans (*Figure 2—figure supplement 1*, *Supplementary file 1* - Table 5). Binary degree did not increase to a greater extent in orphans than non-orphans in either contact or proximity-based networks (*Supplementary file 1* - Table 5).

## Relationship changes following maternal loss

GAMMs were also used to investigate changes in individual pairwise relationships pre- and post-maternal loss. Our first pair of models included all pairwise relationships involving an immature gorilla (both orphans and non-orphans). They demonstrated that affiliative contact with dominant males, subordinate males, and subadult females increased to a greater extent for orphans than other immature gorillas within the same group during the same time period (n = 3486, *Figure 3*; *Supplementary file 1*-Table 6). Based on proximity, orphan's relationships with dominant males, subordinate males, adult females, subadult females, and juveniles strengthened more than those between other immature gorillas and the same age-sex classes of group members (n = 3486, *Figure 3*; *Supplementary file 1* Table 6). Our second pair of models examined only pairwise relationships involving an orphan (n = 755) to provide more detailed information on how orphans relationships changed. The extent to which an orphan's affiliative contact with and proximity to other group members increased following maternal loss did not differ depending on the orphan's sex (*Table 5*). The increase in proximity with other group members following maternal loss was smaller for older orphans but this difference was not significant for affiliative contact (*Table 5*). This suggests that social support after maternal loss through proximity with other group members is lower for older orphans who may already have been less reliant on their mothers. Age-mates (those within 2 years age of the orphan) showed a greater increase in proximity after maternal loss relative to other group members. However, this was not the case for affiliative contact (*Table 5*). The change in relationship strength between maternal siblings (hereafter, siblings) after maternal loss depended on the age-sex class of the sibling (*Figure 3—figure supplement 1*). Subordinate adult males and subadult females had more affiliative contact with younger siblings following maternal loss but siblings in all other age-sex classes did not (*Table 5*). Both forms of social support (affiliative contact and proximity) showed the greatest increase from dominant males (*Figure 3*, *Table 5*). For affiliative contact this was significantly greater than all other age-sex classes, but for proximity the increase was only significantly greater than that of subordinate adult males.

## Changes in the relationship with adult males following maternal loss

For 67% of orphans of known paternity (n = 18), the dominant male at the time of maternal loss was their genetic father. Paternity did not influence the social support provided by adult males after maternal loss (contact: z = 1.130, p=0.262; proximity: z = −0.552, p = 0.583) but adult male maternal siblings increased both their affiliative contact and proximity more than non-siblings (contact: z = 3.807, p < 0.001; proximity: z = 2.237, p = 0.028). However, the increased social support from adult male maternal siblings relative to non-siblings was largely driven by an effect in subordinate males, whilst social support from dominant males did not differ greatly by kin relationship (*Supplementary file 1* - Table 7, *Figure 3—figure supplement 2*).

## Discussion

In contrast to many social mammals with extended periods of mother–offspring co-residence (*Andres et al., 2013*; *Stanton et al., 2020*; *Foster et al., 2012*), in mountain gorillas, we found no evidence for higher mortality in immature offspring of either sex following maternal loss. Whilst our sample size of 59 orphans between the ages of 2 and 8 years may limit our ability to detect relatively weak effects on survival, significant effects have been found in other species with comparable sample sizes (*Stanton et al., 2020*). This suggests that either maternal loss after offspring have reached 2 years of age does not reduce survival in mountain gorillas, or that the reduction in survival is considerably weaker than that observed in other species and therefore undetectable within our sample.

The effect of maternal loss on dispersal decisions differed depending on the age at which maternal loss occurred. Whilst no effect was found for those orphaned as infants, juvenile- or subadult-orphaned gorillas were more than twice as likely to disperse than non-orphans (although for females this effect was not quite significant). This suggests that there may be some benefits to co-residence with mothers for both sexes that do not directly translate to survival, but could influence lifetime reproductive success, like the faster maturation and higher dominance in female chimpanzees (*Walker et al., 2018*; *Foerster et al., 2016*) or greater mating opportunities in male bonobos (*Surbeck et al., 2019*). Alternatively, potential benefits may not be specific to the mother–offspring

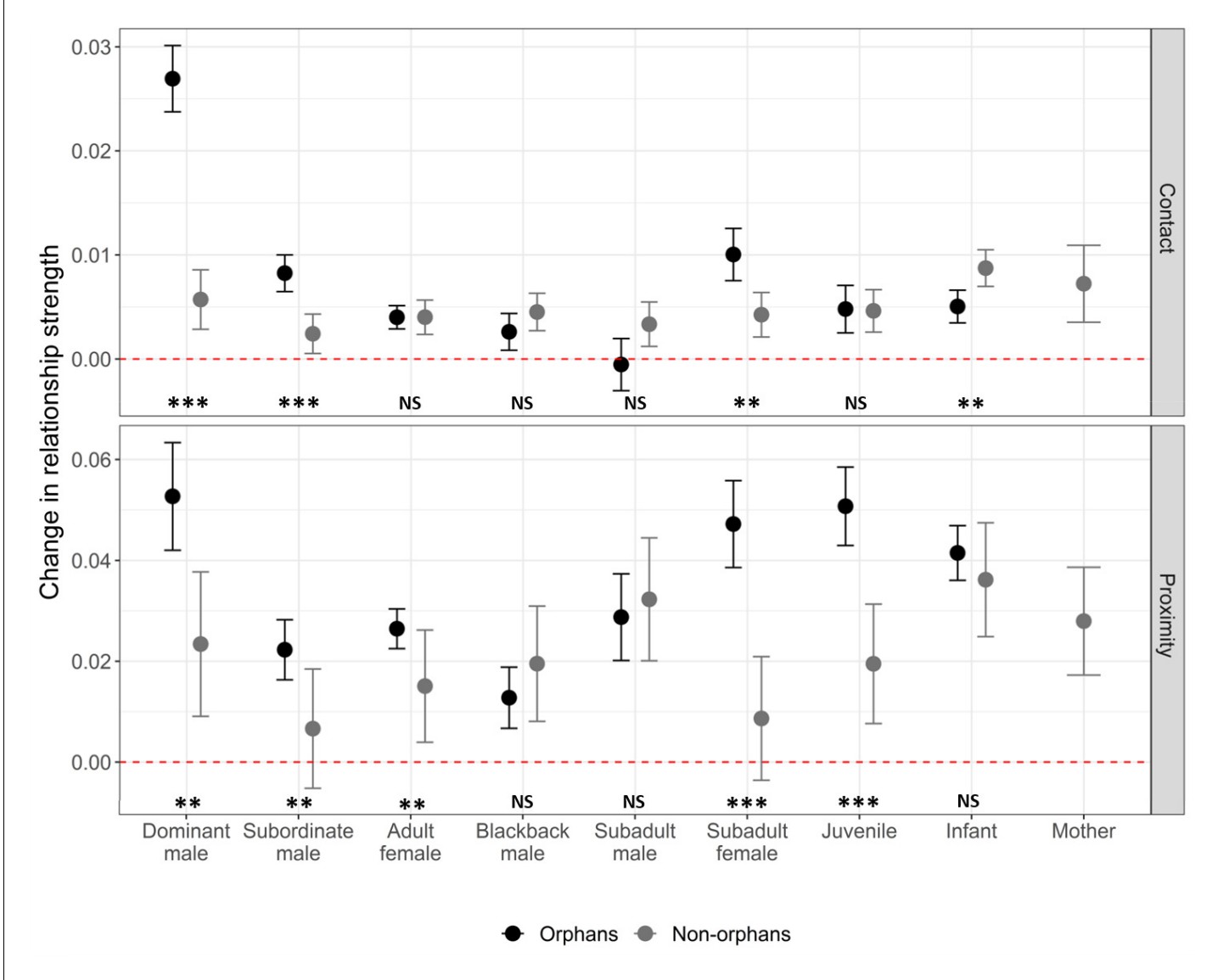

**Figure 3.** The change in relationship strength (SRI) between immature gorillas and other group members in both affiliative contact and proximity, between the 6 months prior to an incident of maternal loss and the 6 months post-maternal loss (n relationships = 3592, orphaned immature gorillas = 31, non-orphaned immature gorillas = 51). Black points show values for orphans, and grey points show values for immature gorillas within the same group that did not suffer maternal loss. Error bars indicate the standard error. Dashed red line indicates no change in relationship strength. NS indicates no significant difference between orphan and non-orphan changes in relationship strength, * indicates significance at <0.05, ** indicates significance at <0.01, and *** indicates significance at <0.001. Exact p-values (L to R), contact: <0.001, 0.001, 0.997, 0.280, 0.122, 0.021, 0.941, 0.019; proximity: 0.006, 0.009, 0.004, 0.265, 0.680, <0.001, <0.001, 0.328. Source data available in file: '*Figure 3—source data 1*'. *Figure 3—figure supplement 1* shows the change in relationship strength between orphans and group members that were their maternal siblings (dark blue) and those that were not (light blue) after maternal loss based on (A) affiliative contact and (B) proximity. *Figure 3—figure supplement 2* shows the change in relationship strength between orphans and dominant and subordinate adult male group members by kinship after maternal loss based on (A) affiliative contact and (B) proximity.

The online version of this article includes the following source data and figure supplement(s) for figure 3:

**Source data 1.** Changes in dyadic relationship strengths following an incident of maternal loss within a gorilla group.

**Figure supplement 1.** The change in relationship strength between orphans and group members that were their maternal siblings (dark blue) and those that were not (light blue) after maternal loss based on (A) affiliative contact and (B) proximity.

**Figure supplement 2.** Change in relationship strength between orphans and dominant and subordinate adult male group members by kinship after maternal loss based on (A) affiliative contact and (B) proximity.

**Table 5.** GAMMs predicting the change in dyadic relationship strength (SRI values for affiliative contact and proximity) between orphans (O) and other group members (GM) following maternal loss (n = 755).

| | Affiliative contact | | | Proximity | | |
|---|---|---|---|---|---|---|
| | Est ± SE | Z | p | Est ± SE | Z | p |
| Intercept | 0.028 ± 0.005 | 5.508 | <0.001 | 0.058 ± 0.018 | 3.180 | 0.002 |
| *Orphan* | | | | | | |
| Age (years) | −0.001 ± 0.000 | −1.640 | 0.101 | −0.004 ± 0.002 | −2.033 | 0.042 |
| Sex (male) | 0.000 ± 0.002 | −0.034 | 0.973 | −0.003 ± 0.006 | −0.503 | 0.615 |
| *Group member* | | | | | | |
| Age-mate (within 2 years) | −0.001 ± 0.002 | −0.623 | 0.534 | 0.020 ± 0.007 | 2.885 | 0.004 |
| Maternal sibling | −0.014 ± 0.011 | −1.305 | 0.192 | 0.010 ± 0.037 | 0.284 | 0.777 |
| *Group member age/sex class (relative to a dominant male)* | | | | | | |
| Adult male (subordinate) | −0.023 ± 0.005 | −4.911 | <0.001 | −0.033 ± 0.017 | −1.973 | 0.049 |
| Adult female | −0.024 ± 0.005 | −5.083 | <0.001 | −0.016 ± 0.017 | −0.959 | 0.338 |
| Blackback male | −0.023 ± 0.005 | −4.731 | <0.001 | −0.032 ± 0.018 | −1.782 | 0.075 |
| Subadult male | −0.026 ± 0.005 | −4.801 | <0.001 | −0.021 ± 0.020 | −1.038 | 0.299 |
| Subadult female | −0.018 ± 0.005 | −3.205 | 0.001 | −0.003 ± 0.020 | −0.151 | 0.880 |
| Juvenile | −0.021 ± 0.005 | −3.930 | <0.001 | −0.002 ± 0.019 | −0.124 | 0.902 |
| Infant | −0.021 ± 0.005 | −4.224 | <0.001 | −0.006 ± 0.018 | −0.334 | 0.738 |
| *Age/sex class–sibling interaction (relative to a non-sibling dominant male)* | | | | | | |
| Adult male (sub.) sibling | 0.041 ± 0.013 | 3.210 | 0.001 | 0.052 ± 0.043 | 1.231 | 0.219 |
| Adult female sibling | 0.015 ± 0.014 | 1.016 | 0.310 | −0.048 ± 0.048 | −0.986 | 0.324 |
| Blackback male sibling | 0.004 ± 0.015 | 0.243 | 0.808 | −0.044 ± 0.049 | −0.896 | 0.371 |
| Subadult male sibling | 0.001 ± 0.017 | 0.042 | 0.966 | −0.107 ± 0.057 | −1.876 | 0.061 |
| Subadult female sibling | 0.058 ± 0.017 | 3.387 | 0.001 | 0.011 ± 0.058 | 0.197 | 0.844 |
| Juvenile sibling | −0.020 ± 0.021 | −0.955 | 0.340 | −0.118 ± 0.072 | −1.643 | 0.101 |
| Infant sibling | −0.004 ± 0.015 | −0.282 | 0.778 | −0.096 ± 0.052 | −1.850 | 0.065 |

Smooth term: s(mean focal scans) Contact: F = 6.955, p=0.008; Proximity: F = 7.288, p<0.001.

The online version of this article includes the following source data for Table 5:

**Source data 1.** Model: relationship change ~ age_O + sex_O + age mate + age sex class_GM * sibling + s(MeanDen), random = ~(Group|ID_O) + (1| ID_GM).

bond. Instead, the loss of a key social partner close to dispersal age may reduce the social benefits of remaining in a group. Without this strong social bond in the natal group, dispersal to a new group may lead to fewer social costs relative to remaining, whilst also providing considerable benefits for inbreeding avoidance (*Vigilant et al., 2015*). This might better explain the age-dependent influence of maternal loss, as by the time those suffering maternal loss as infants approach an age at which dispersal could occur, they may have compensated for the loss of such a key social partner through strengthening their relationships with other group members. Our analyses also suggest that the strengthening of relationships with group members post-maternal loss may be greatest for younger individuals, which could further reduce the likelihood that they disperse later in life.

We found no significant effect of maternal loss on the elements of female reproduction we investigated. Female orphans gave birth slightly younger, but despite this, their first offspring was marginally more likely to survive infancy. Whilst neither effect was significant, the direction of the effect, with earlier first birth, is consistent with the stress acceleration hypothesis: that those suffering from early caregiving adversity may show accelerated development as an adaptive strategy to low parental care (*Ellis, 2004*; *Callaghan and Tottenham, 2016*). Female dispersal has also not been found to cause any reproductive delays in mountain gorillas, suggesting that the potentially higher likelihood of those suffering maternal loss as juveniles or subadults to disperse is unlikely to reduce their reproductive capacity (*Robbins et al., 2009b*).

Male reproduction was harder to assess due to limited paternity data and later sexual maturity. However, as dominant males usually sire the majority of offspring, even within multi-male groups (*Nsubuga et al., 2008*; *Stoinski, 2009b*; *Bradley et al., 2005*), analyses of male dominance status provided some insights into the reproductive success of male orphans. Although only around half of males ever reached dominant male status, some males in both orphan age categories were able to do so, demonstrating that this is possible in the absence of maternal support. Small sample sizes limited the extent of analysis possible, but maternal loss appeared to have differing effects depending on the age of maternal loss, with infant-orphaned males more likely and juvenile- or subadult-orphaned males less likely to become the dominant male of a group relative to males that did not suffer maternal loss. This mirrors the effect of maternal loss on dispersal decisions. Modelling has suggested that males that disperse suffer a 50% reduction in lifetime reproductive success (*Robbins and Robbins, 2005*), as they lose the potential for mating opportunities as a subordinate male within their natal group or for eventually taking over as dominant male of that group and must instead attempt to attract females to form a group of their own (*Stoinski et al., 2009a*). It is therefore likely that males suffering maternal loss as juveniles or subadults may suffer from reduced siring opportunities throughout their lives due to their increased likelihood of dispersal. As discussed above, it is possible that the greater strengthening of relationships between group members and those orphaned at an earlier age may reduce the likelihood of these infant-orphaned gorillas dispersing relative to those suffering maternal loss as juveniles or subadults, ultimately influencing male orphans' subsequent reproductive success.

Overall, these findings suggest there are no strong negative fitness consequences for maternal loss in female mountain gorillas over 2 years of age, although we cannot rule out longer-term effects on lifetime reproduction. In males, maternal loss after 2 years does not appear to influence survival but maternal loss as juveniles or subadults (4–8 years) could lower their future reproductive success. This lack of effect on survival reflects patterns found in human populations with natural fertility and mortality, where the negative effect of maternal loss on survival declines substantially with age and in many cases disappears entirely over the age of 2 (*Sear and Mace, 2008*). However, it is contrary to research in most other social mammals with long periods of mother–offspring co-residency (*Snyder-Mackler et al., 2020*; *Andres et al., 2013*; *Stanton et al., 2020*; *East et al., 2009*; *Tung et al., 2016*; *Watts et al., 2009*; *Nakamura et al., 2014*; *Goldenberg and Wittemyer, 2017*; *Goldenberg and Wittemyer, 2018*). In these species, one of the major mechanisms posited to link maternal loss with increased mortality is through the social adversity caused by mother absence. Specifically, mother absence is found to reduce social integration (*Tung et al., 2016*; *Archie et al., 2014*), competitivity (*Surbeck et al., 2019*; *Samuni et al., 2020*), dominance rank (*Strauss et al., 2020*; *East et al., 2009*; *Maestripieri and Mateo, 2009*; *Lea et al., 2014*), and opportunities for social learning (*Stanton et al., 2020*; *Foley et al., 2008*; *Brent et al., 2015*; *Estienne et al., 2019*). But the lack of increased mortality in gorillas and certain human populations suggests that mother absence does not always lead to social adversity in mammals with extended maternal care.

When immature mountain gorillas suffered maternal loss, we found that their social relationships with other group members estimated through proximity increased in strength (weighted degree), and their social integration within the group (eigenvector centrality) increased considerably. There was no increase in binary degree suggesting that orphans strengthen existing social bonds rather than forming new ones following maternal loss. The increase in orphan eigenvector centrality was especially large (*Figure 2*), which may be explained by the strengthening of the orphan-dominant male bond in particular, with dominant males typically being extremely well connected within the group network. The increase in proximity-based relationship strength with other group members was actually great enough to outweigh the loss of their mother, although this was not the case for relationships based on contact which did not change significantly more than non-orphans overall. In contrast to fission–fusion social systems such as in chimpanzees, where group composition regularly changes, mountain gorilla orphans remain in stable cohesive social groups in which their social integration is not dependent on their mother – as shown by the strengthening of relationships following maternal loss. Their increased social integration within the larger cohesive group immediately following maternal loss suggests that mountain gorilla orphans are unlikely to have fewer opportunities for social learning as they are regularly in close proximity to multiple group members. Gorillas also do not appear to require the complex feeding techniques such as nut-cracking or termite-fishing, for which close contact with mothers may be most beneficial (*Lonsdorf et al., 2004*; *Estienne et al.,*

*2019*). Instead, common group-membership may be sufficient for learning the complex foraging behaviours observed in gorillas (*Watts, 1984*; *Byrne et al., 2001*) as groups travel and feed as a cohesive unit.

The greatest increases in relationship strength following maternal loss were found with the dominant male of the group, regardless of whether or not he was the genetic father. This suggests that unlike in elephants where maternal loss leads to weaker relationships with dominant group members, gorillas that suffer maternal loss instead have increased access to the most dominant member of their group. Whilst the cohesion of gorilla social groups already limits the extent to which orphans are likely to suffer from reduced access to resources or knowledge, the strengthening of social relationships, particularly with the dominant male, is likely to further buffer any potential reduction to their competitivity or future dominance rank within the group. Other cohesive social groups in which the effects of maternal loss have been studied have primarily been matriarchal, with groups led by older females, e.g. elephants (*Goldenberg and Wittemyer, 2017*; *Goldenberg and Wittemyer, 2018*), or where strong female dominance hierarchies are inherited, e.g. spotted hyenas (*East et al., 2009*; *Watts et al., 2009*) and cercopithecine species such as the savannah baboon (*Archie et al., 2014*). In contrast, males are the dominant sex in gorillas (*Wright et al., 2019*). Female dominance hierarchies are relatively weak, and the common dispersal of both sexes means maternal support is unlikely to provide considerable benefits for dominance to offspring of either sex. Orphaned gorillas are therefore unlikely to suffer costs in this regard. However, for males that remain in their natal group it is possible that stronger relationships with the dominant male, such as those developed post-maternal loss, could aid dominance acquisition. For males that lose their mothers at particularly early ages and show some of the strongest relationships with the dominant male as a result, it is possible that this could ultimately improve their chances of inheriting dominance of their natal group (*Stoinski et al., 2009a*; *Robbins and Robbins, 2005*). Although, due to the rarity of such events (three cases in 53 years) there may never be a large enough sample size to thoroughly investigate such a hypothesis.

The lack of paternity discrimination in the support provided by adult males after maternal loss is consistent with previous findings on paternal care in mountain gorillas, where the highest ranking males sire the majority of offspring and provide the most care, regardless of paternity (*Rosenbaum et al., 2018*; *Rosenbaum et al., 2015*). As observed in chimpanzees (*Reddy and Mitani, 2019*), support from some maternal siblings also appears to occur, with relationships between subordinate adult males and subadult females, and their younger siblings strengthening following maternal loss. In these two classes, caring for siblings may have additional benefits to those of inclusive fitness (*Riedman, 1982*). Care of infants by adult male mountain gorillas has been found to be linked with dramatically higher reproductive success. Males in the top tertile for showing affiliative behaviour towards infants were found to sire 5.5 times more infants than those in the bottom tertile, even after accounting for rank (*Rosenbaum et al., 2018*). This suggests that females may prefer males that demonstrate more caring behaviour towards infants, and that subordinate male kin that increase their care of younger maternal siblings following maternal loss may additionally increase their own reproductive success. Pre-reproductive female siblings, particularly subadult females, may also benefit through developing parental experience that may improve their own reproductive success (*Riedman, 1982*).

Our analyses of social relationship changes used undirected data on pairwise proximity and affiliative contact. Whilst both measures of affiliation indicate at a minimum, high tolerance for the other individual, we cannot determine who was responsible for initiating the relationship changes. Future research could investigate whether it is the orphans themselves that are strengthening their relationships by approaching other group members more often or whether it is the other group members responding to the maternal loss faced by these young gorillas. Our study also only investigates the immediate social response to maternal loss in the 6 months after the mother dies or leaves the group. Whilst this might be expected to be the period where individuals face the greatest social costs from maternal loss, our analyses cannot tell us whether the social buffering we detected during this period is maintained in the longer term. However, previous observations have suggested that the close associations that young gorillas develop following maternal loss continue into subadulthood (*Robbins et al., 2005*). It is therefore highly likely that the dramatic short-term changes we have detected are part of a longer-term social response to maternal loss with the potential to overcome the long-term fitness costs of maternal loss.

In many human populations, care from other family members is believed to buffer the negative consequences of maternal loss, but the identity of these carers can vary greatly between populations. In rural Gambia, elder sisters and maternal grandmothers increased offspring survival but not fathers, other grandparents, or brothers (*Sear et al., 2003*). In contrast, in the Ache of Paraguay the loss of fathers significantly impacted offspring survival but the loss of grandparents or adult siblings did not (*Hill and Hurtado, 2017*). Support from family members in rearing offspring is thought to be a human universal but the composition of those families and the specific family members involved in cooperative care appears to be flexible and responsive to ecological conditions (*Sear and Mace, 2008*). In killer whales, as observed in many human populations, grandmothers have been found to influence offspring survival and this additional support could in part contribute to the lack of reduced survival for younger female killer whales after maternal loss (*Foster et al., 2012*; *Nattrass et al., 2019*). In mountain gorillas, grandparents and their grand-offspring are rarely in the same social group, as females often transfer between groups multiple times within their lives (*Robbins et al., 2009a*). Instead, fathers (or dominant males with a high likelihood of paternity), siblings, and group members close in age seem to play a key role in buffering the detrimental effects of maternal loss. What appears to set humans, gorillas, and possibly killer whales apart from other species with extended maternal care where high fitness costs to maternal loss are observed is the potential for cooperative care from within the social group.

## Conclusion

We found that immature mountain gorillas do not appear to face increased social adversity or a detectable reduction in fitness following maternal loss. It is not yet possible to demonstrate the direct link between the strengthening of relationships with other group members after maternal loss and the absence of fitness costs to maternal loss. However, our analyses show that at least in the short-term, a key mechanism by which maternal loss is hypothesized to lead to reduced survival and fitness in other social species – social adversity - does not apply in mountain gorillas over the age of 2 years. The social support provided by other group members within mountain gorillas' cohesive social groups, particularly from dominant males, siblings, and those close in age, appears to buffer against the negative consequences of maternal loss. In mountain gorillas, like humans (*Sear and Mace, 2008*; *Lahdenperä et al., 2004*; *Sear et al., 2000*; *Hurtado and Hill, 1992*; *Andersson et al., 1996*; *Campbell and Lee, 2002*; *Bentley and Mace, 2009*) social support appears to come from a number of group or family members. This could provide a buffer to the loss of any single relationship, even one as important as the mother–offspring relationship, once an individual can be nutritionally independent. In the absence of nepotistic matriarchal dominance hierarchies and when social buffering is possible due to cohesive strongly bonded social groups, it may matter less who is providing care as long as care is provided.

## Materials and methods

### Demographic data

Mountain gorillas in the Volcanoes National Park, Rwanda, have been monitored almost continuously by the Dian Fossey Gorilla Fund's Karisoke Research Center since 1967. Habituated mountain gorilla groups are monitored daily by field teams who collect data on demography, behaviour, ranging, and health. From 1967 to 2015 (inclusive), 59 (28 males, 31 females) out of the total 200 immature mountain gorillas (102 males, 98 females) that reached the age of at least 2 years suffered maternal loss between the ages of 2 and 8 through the death or permanent transfer of their mother.

Gorillas were classified as infants up to 4 years of age, as juveniles from 4 to 6 years of age and as subadults from 6 to 8 years of age (*Breuer et al., 2009*). From 8 years of age females were classified as adults. Males were classified as blackbacks from 8 to 12 years. From 12 years, males were classified as either subordinate adult males or dominant adult males from their dominance hierarchy. Male dominance hierarchies were based on displacements and avoidances using the Elo-rating method (*Albers and de Vries, 2001*; *Neumann et al., 2011*). Dominance hierarchies were calculated using the R package EloRating, version 0.43 (*Neumann and Lars, 2014*) as described by *Wright et al., 2019*. Only one adult male was classified as dominant in each group at a given time.

Dominant males were those with the highest dominance status unless they were the only adult male in the group in which case they were automatically classified as dominant.

## Survival

The youngest infant to survive maternal loss was a 2.45-year-old female. Our data set included only one infant younger than this that suffered maternal loss at 0.67 years and died after 1 day. Three infants aged 1.91, 2.42, and 2.52 became separated from their mothers after a suspected poacher encounter. During this separation they travelled with a small number of group members not including their mothers. The 1.91-year-old died after 6 days of separation. The 2.42-year-old died after 9 days of separation. The 2.52-year-old survived until they were reunited with their mother and the larger group 18 days after the initial separation. These infants were not considered as orphans in the data set as their mothers did not permanently transfer or die, but in combination with those that did, they suggest that infant mountain gorillas cannot survive independently from their mothers under the age of at least 2. We therefore investigated the effect of maternal loss after the age of 2.

To determine the effect of maternal loss on survival, we carried out a Cox-proportional hazards analysis separating individuals over the age of 2, based on four general age classes of maternal loss: (a) infants, (b) juveniles, (c) subadults, and (d) non-orphans, where mothers did not die or leave the group before the individual reached 8 years of age. First, we ran a Cox-proportional hazards model for each sex (males: n = 102, females: n = 98) with a time-varying covariate for the age at maternal loss and using the four classes as covariates. Due to the small sample size for the subadult class, we merged this with the juvenile class into a single juvenile/subadult class and ran a new set of Cox-proportional hazards models on these new classes. In all cases we truncated the analysis to start at age 2.

In order to verify our results and to account for the uncertainty in some of the dates of birth, we ran a Bayesian survival trajectory analysis (*Colchero and Clark, 2012*; *Colchero et al., 2012*) for each sex truncated at the age of maternal loss for all orphans, and at 2 years of age for non-orphans. We used orphan status as a binary covariate (orphan vs non-orphan) and, using the *Siler, 1979* mortality model for the baseline mortality, we tested three models: (a) no covariates (i.e. null model where all individuals have the same hazard rate); (b) proportional hazards (where mortality differs proportionally between orphan classes); (c) covariates modifying all Siler mortality parameters (where each orphan class has a different age-specific mortality). We used deviance information criterion for model fit and selection (*Spiegelhalter et al., 2002*; *Celeux et al., 2006*). Model (b) is equivalent to the Cox-proportional hazards model. However, these tests facilitate further exploration of the hypotheses on the effect of maternal loss on mortality, namely that there is no effect (model a) or that the entire age-specific trajectory of mortality changes for each category.

## Female dispersal and reproduction

Between 1967 and 2019, 66 females gave birth to what was known to be their first offspring. For 53 of these females, their age could be accurately estimated within a 90-day period. For these individuals, we extracted their age at first birth, whether they dispersed from their natal group prior to first birth and whether they had suffered maternal loss when immature, from the long-term database. Maternal loss was investigated with the orphan age classes described above with juvenile and subadult classes merged due to small sample sizes. We investigated the effect of maternal loss on the decision of females to disperse from their natal group prior to their first birth using a binomial generalized linear model. We investigated the effect of both maternal loss and dispersal prior to first birth on age at first birth using a generalized linear model with a Gaussian distribution. Due to the positive skew of age at first birth, we used the square root of age at first birth minus 8 (the earliest recorded age at first birth) as the response variable. We checked Q–Q plots to verify the normal distribution of residuals and ran Levene tests using the 'rstatix' package to check for heteroskedasticity. Finally, we examined survival of each female's first offspring through infancy (1: survived to 4 years, 0: died before reaching 4 years) according to age at first birth, dispersal prior to first birth (1: yes, 0: no), and maternal loss, with merged juvenile and subadult classes as above, using a binomial generalized linear model. Multicollinearity of all models with multiple variables was checked using variance inflation factors in the 'car' R package.

## Male dispersal and dominance

Between 1967 and 2019, 56 males whose age could be accurately estimated within a 90-day period reached the median age of dispersal (16 years). We used a binomial generalized linear model to predict whether each of these males dispersed from their natal group prior to this age according to orphan age classes (as above). To investigate the effect of maternal loss on dominance, we gave males that became the dominant male of a stable group for at least six consecutive months a dominance score of 1. This included adult males of single-male groups and the most dominant male of multi-male groups based on Elo-ratings. Males that never reached dominance or only transiently (for <6 months) received a score of 0. We recorded the age at which a male first became dominant for those for which this could be accurately estimated within a 90-day period. This represented the age at which they first successfully attracted and retained a female to join their group, the age at which they split from their natal group with at least one adult female to form a new group, or the age at which their Elo-rating surpassed that of all other adult males in their group, if those groups remained independent and did not disintegrate within 6 months of that date. The mean age (± standard deviation) at which a male reached dominance was 17.88 ± 2.56 years. The median was 17.29 years. The oldest age at which a male first became dominant was 22.99 years. Therefore, to investigate the influence of maternal loss on dominance status we analysed only males that had survived and remained in the study population until at least 23 years (n = 40). We did not attempt to statistically examine the effects of maternal loss on dominance status due to small sample sizes.

## Social network analysis

Habituated gorilla groups were monitored for up to 4 hrs daily and all gorillas were individually identified by physical characteristics. Behavioural data were collected on each group member via 50 min focal sampling during which the researcher would typically be within 10–20 m of the focal individual. Researchers systematically worked their way through a randomly ordered list of all individuals in the group. If an individual could not be observed (e.g. obscured by dense vegetation), the researcher moved on to the next individual on the list and returned to them the subsequent day. During focal sampling, a focal scan was completed every 10 min which recorded all gorillas within 2 m of the focal individual and all gorillas in physical contact with the focal individual. This data was therefore in the form of frequencies (the number of focal scans during which individuals were associating) rather than durations. Affiliative contact included resting, playing, or feeding in contact and grooming, and excluded physical aggression. This focal sampling approach limited the extent of potential sampling bias due to individual-level differences in observation propensity, with all group members systematically observed and an extremely low likelihood of individuals within 2 m or in physical contact of the focal individual being missed at these close proximities.

The social response of group members to an incident of maternal loss was investigated in 31 of the 59 total cases – those that suffered maternal loss after 2003 for which adequate social behaviour data was available (more than 12 focal scans of the individual were recorded in the 6 months prior to maternal loss and the 6 months after maternal loss, *Supplementary file 1* - Table 3). For each case of maternal loss, two types of weighted social network were constructed based on (a) 2 m proximity and (b) affiliative contact. For each type, edge values of the networks were calculated using the Simple Ratio Index (SRI) (*Whitehead, 2008*) with edges between a pair of individuals calculated as the proportion of focal scans of either individual during which the pair was recorded as associating. These values represented an estimate of the proportion of time two individuals were either within 2 m of each other or in physical contact. For example, in the contact network a value of 1 would indicate that the two individuals were in physical contact every time a focal scan of either individual was conducted, whilst 0 would indicate that they were never observed in physical contact during a focal scan. Both network types were constructed for two time periods for each case of maternal loss: pre-maternal loss (the 6 months leading up to maternal loss) and post-maternal loss (the 6 months immediately after maternal loss). Social networks were constructed using all focal scans during these time periods. Only gorillas for which more than 12 focal scans were available in both of the 6 month periods (pre- and post-maternal loss) were included in the networks (*Farine and Whitehead, 2015*). This meant that only the social relationships with group members that were present both pre- and post-maternal loss were analysed, except for the mother–offspring relationships pre-maternal loss which

were extracted separately. The mean (± SD) number of focal scans used to estimate edge values pre-maternal loss was 144.81 ± 90.10 and post-maternal loss was 149.16 ± 88.56.

## Changes in network position following maternal loss

Binary degree, weighted degree, and eigenvector centrality in the pre- and post-maternal loss networks (excluding orphan's mothers) were calculated using the 'igraph' package for both network types (contact and proximity). To enable comparison across networks of the same type, binary degree and weighted degree metrics were calculated as a proportion of the maximum value for an individual within the network (as is already the case for eigenvector centrality). This meant that for all networks, each metric could have a maximum value of 1 and a minimum value of 0. Metrics were extracted for all immature gorillas that were between the ages of 2 and 8 years on the date that an immature individual within their group suffered maternal loss. We then calculated the change in these network metrics between time periods for all immature gorillas in each set of group networks and used GAMMs in the 'gamm4' R package to determine whether orphan's network metrics changed differently to those of non-orphans within the same group during the same time period. To ensure significant changes were not driven by unusually high or low initial values, the deviance of the network metric in the initial network from the mean value for immatures in the network was calculated. Orphan status, age, and the deviance of the initial value from the group mean were included in the model as fixed factors. The number of focal scans per individual across both time periods was included as a smoothing factor to account for any potential differences driven by sampling effort. The specific set of group networks (the pair of networks for each incident of maternal loss) was included as a random effect to account for differences in network composition (*Supplementary file 1* - Table 5).

Due to the non-independence of network metrics, null models generated through node-level permutations were used to assess the significance of orphan status on the change in network metric. Permutations were run by swapping orphan status between immature gorillas (orphans and non-orphans) within the same set of paired group networks (same maternal loss incident). Orphans that suffered maternal loss in groups with no other non-orphaned immature gorillas were excluded from the analyses, leaving a sample of 19 incidents of maternal loss, 28 orphans, and 108 non-orphans (*Supplementary file 1* - Table 3). The mean age ± SD of orphans was 5.12 ± 1.49 years and for non-orphans was 4.71 ± 1.71 years. 10,000 sets of node-permutations were generated by permuting orphan status between immature gorillas in the same network and extracting node labels every 200th permutation. The same GAMMs were then run on all 10,000 sets of node permutations to produce a null distribution of t-values for the effect of orphan status. $P_{null}$ was calculated using a two-tailed approach. For observed t-values greater than the median of the null distribution, $P_{null}$ was calculated as:

$$2 \times \frac{\text{number of null t} - \text{values greater than observed t} - \text{value}}{\text{total null t} - \text{values}}$$

For observed t-values lower than the median of the null distribution, $P_{null}$ was calculated as:

$$2 \times \frac{\text{number of null t} - \text{values lower than observed t} - \text{value}}{\text{total null t} - \text{values}}$$

## Relationship changes following maternal loss

We extracted SRI edge values (representing the strength of relationship between a pair of gorillas) between the orphan and all other group members pre- and post-maternal loss. We also extracted SRI edge values between all immature gorillas (aged 2–8 years) within the same group that had not suffered maternal loss and all other group members, for the same time periods. We then calculated the change in relationship strength between an immature gorilla (both orphans and non-orphans) and other group members following an incident of maternal loss within the group as the change in SRI value between periods (SRI post-maternal loss – SRI pre-maternal loss) in both network types (contact and proximity). GAMMs were used to predict this change for both affiliative contact and proximity which enabled the non-independence of relationships involving the same individual to be accounted for through random effect structures and sampling variation to be accounted for using a smoothing term. We ran an initial set of GAMMs on the change in SRI values for relationships

involving the orphans or other immature gorillas within the same group that had not suffered maternal loss, excluding mother–offspring relationships. This was to verify that changes in the relationships of orphans with other group members were significantly different to those observed for other immature gorillas present within the same group during that period. These GAMMs included the identity of the immature gorilla as a random factor, nested within the specific maternal loss incident (group and time period), nested within the group. The identity of the other group member was included as a further random factor (*Supplementary file 1* - Table 6). The mean number of focal scans used to estimate the SRI value across both time periods was included as a smoothing term in the model to account for any biases from sampling intensity. The age and sex of the immature gorilla were included as fixed factors, along with the age-sex class of the other group member, whether the immature gorilla suffered maternal loss during this period (1: Yes, 0: No), and the interaction between maternal loss and the age-sex class of the other group member. Non-orphan relationships where the other group member was an orphan were excluded from the analysis.

We then ran GAMMs on only the relationships involving orphans to investigate in more detail how these changed following maternal loss. The identity of the orphan was included as a random factor, nested within the group. The identity of the other group member was included as a further random factor (*Table 5*). The mean number of focal scans used to estimate the SRI value across both time periods was again included as a smoothing term. The age and sex of the orphan were included as fixed factors to predict the social response of group members, as well as whether those group members were maternal siblings (1: Yes, 0: No) and whether those group members were age-mates (1: <2 years age difference with the orphan, 0: ≥2 years age difference). This 2-year cut-off was chosen to be consistent with the width of age categories, such that age-mates would be within the same age class during large proportions of their immature life. The age-sex class of the other group member at the date of maternal loss and the interaction between this and maternal sibling status were also included as fixed factors for predicting the change in relationship.

## Adult male relationship changes following maternal loss

Paternity was known for 18 of the 31 orphans for which social data was available, from a previous study (*Vigilant et al., 2015*). To investigate the influence of paternity, we ran an additional set of GAMMs to predict the change in relationship (both contact and proximity) between adult males and orphans following maternal loss (n = 121). The identity of the orphan and the identity of the adult male were included as random factors (*Supplementary file 1* - Table 7). The mean number of focal scans was included as a smoothing term. Orphan age, orphan sex, male dominance, and the binary kinship variables: maternal sibling and paternity were included as fixed factors. The interaction between dominance and sibling status was also included in the model, but the interaction between dominance and paternity could not be, due to possible issues of multicollinearity (variance inflation factors > 3).

## Acknowledgements

We are grateful to the Rwandan Development Board (RDB) for their long-term support of the Karisoke Research Center and thank the Karisoke field staff for collection of the data. We are thankful to Edward Wright for his support with the Elo-rating analysis for male dominance hierarchies. We are grateful to Lauren Brent and her research group for providing insightful feedback and discussion on the project, and to members of the University of Exeter's Social Network Club for their advice on statistical analyses. We thank Elizabeth Lonsdorf for her valuable feedback on the manuscript and the editors and reviewers at eLife for their constructive comments.

## Additional information

### Funding

No external funding was received for this work.

## Author contributions
Robin E Morrison, Data curation, Formal analysis, Visualization, Writing - original draft, Writing - review and editing; Winnie Eckardt, Conceptualization, Data curation, Formal analysis, Project administration, Writing - review and editing; Fernando Colchero, Formal analysis, Visualization, Writing - review and editing; Veronica Vecellio, Data curation, Project administration; Tara S Stoinski, Conceptualization, Funding acquisition, Project administration, Writing - review and editing

## Author ORCIDs
Robin E Morrison (ID) https://orcid.org/0000-0001-9161-4734

## Ethics
Animal experimentation: The research presented here was non-invasive and did not involve any animal experimentation. It was approved by the Rwandan Development Board and conducted in accordance with the ethical standards of the Dian Fossey Gorilla Fund and the International Primatological Society's Code of Best Practices for Field Primatology.

## Decision letter and Author response
Decision letter https://doi.org/10.7554/eLife.62939.sa1
Author response https://doi.org/10.7554/eLife.62939.sa2

# Additional files

## Supplementary files
• Supplementary file 1. Supplementary Tables 1–7 including analyses with merged juvenile and sub-adult orphan classes, information on group composition, and sampling at each incident of maternal loss and GAMMs predicting the change in both orphan and non-orphan network position and social relationships.

• Transparent reporting form

## Data availability
Demographic data available within the Primate Life Histories Database at https://datadryad.org/stash/dataset/doi:10.5061/dryad.v28t5 Social data available as source data files.

The following previously published datasets were used:

| Author(s) | Year | Dataset title | Dataset URL | Database and Identifier |
|---|---|---|---|---|
| Bronikowski, Anne M | 2017 | Data from: Female and male life tables for seven wild primate species | https://doi.org/10.5061/dryad.v28t5 | Dryad Digital Repository, 10.5061/dryad.v28t5 |

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
