## [Decision Letter]

**Acceptance summary:**

We were very pleased to see that the previous round of recommendations and suggestions were taken onboard and responded to in full. We believe the manuscript reads very clearly, and the methods are well explained throughout. As expressed in previous decision letters, we are highly impressed by this interesting work and believe it will be a valuable contribution to the literature that will be enjoyed by *eLife*'s broad readership.

**Decision letter after peer review:**

Thank you for submitting your article "Social groups buffer maternal loss in mountain gorillas" for consideration by *eLife*. Your article has been reviewed by Christian Rutz as the Senior Editor, a Guest Editor, and three reviewers. The reviewers have opted to remain anonymous.

The reviewers have had the opportunity to discuss their reviews with one another, and the Guest Editor has drafted this decision letter to help you prepare a revised submission.

We would like to draw your attention to changes in our revision policy that we have made in response to COVID-19 (https://elifesciences.org/articles/57162). Specifically, we are asking editors to quickly process manuscripts, like yours, that they judge can potentially stand as *eLife* papers without additional data. Thus, the revisions requested below are aimed at: (a) addressing clarity and presentation of the manuscript; and (b) addressing further considerations about the current analyses and arising conclusions.

Summary:

This manuscript provides an in-depth assessment of the consequences of maternal loss for wild mountain gorillas, and its rigorous and detailed approach was appreciated by all reviewers. However, some concerns were raised by the reviewers in relation to the reported analyses, and the wording of certain text passages.

Essential revisions:

We judge that the manuscript would benefit from addressing the reviewers' comments point-by-point, and have therefore decided to append their full reports below. In relation to some of the comments detailed below, we would like to highlight the following three points:

1) The use of paired t-tests for the examination of social changes is a bit confusing. Specifically, we are unsure how these properly control for repeated sampling of individuals when considering changes in dyadic association scores. Also, the changes reported here appear to just assess how orphan behaviour altered between the pre- and the post-maternal loss period, but how do we know that such changes aren't expected anyway over time? Ideally, these analyses would directly compare orphaned to non-orphaned individuals. Note the comments from reviewer #1 about these paired t-tests that also need to be addressed.

2) Animal social networks (either dyadic association scores or individual-level metrics) are strongly influenced by observation. But it is not clear how these analyses controlled for, or considered, how individual-level differences in observation propensity might alter inferences of their social network position/association scores, and what this might mean for the arising conclusions. We think the manuscript would benefit from including a much more detailed report of how individuals differed in their likelihood of being observed in relation to the social behaviour considered here, and how this may shape the arising networks, and whether controlling for these effects might alter the results/conclusions.

3) We are unsure why only weighted degree was used as a network metric; reviewer #3 also raises a point about this metric. We suggest that this could be addressed through also considering mean average non-zero dyadic bond scores as a metric. Furthermore, you may want to re-run the network metric-based analyses, but with other measures of social behaviour (e.g., eigenvector centrality) to get an idea of how wider indirect centrality within the network may relate to maternal loss.

Reviewer #1:

This is an interesting paper about how the loss of a mother influences individuals future development, or rather the fact that it might not due to social connections with other group members. The paper examines a very comprehensive set of question, and on the whole I found it very well written, though I have some quibbles about how certain ideas are presented/supported. The dataset presented seems impressive. The methods used generally seem appropriate and robust, though I'd like to see some extra justifications for some of the approaches taken during network analysis. For more info, see detailed comments:

Introduction

I think an example of how the presence of the mother affects fitness in later life would be nice here, in order to lend weight to the statement about maternal loss having affects throughout an individual's lifetime. The third paragraph contains examples, and currently feels repetitive in terms of the points raised.

Similarly, I think "these negative changes" could be expanded on in the preceding sentences with an example of negative changes to the social environment.

I am not sure how convinced I am that the human example here adds to the authors' point. As mentioned above, I would prefer to see more general examples of these ideas/effects in social mammals (or animals in general). If the authors' really want to keep a human example, I'd like to see significantly more detail about the circumstances in which this result was obtained.

It feels slightly odd to discuss sex specific consequences before the general consequences.

As above, I am slightly unconvinced by the need/relevance to relate these results back to humans. This might be a result of me not having an anthropology background of course. Related to this, I feel this is making a rather large assumption about the familiarity a reader might have with this literature. Some small details about what these populations are might help convince that they are relevant to build the argument in the same paragraph as killer whales. There are a few other instances of this in the Discussion too, but I'll avoid repeating myself.

Any directional predictions going into the study? There are quite a few variables under investigation, and it would be nice to link them more strongly to the ideas articulated in the Introduction. Is social buffering more likely in certain group compositions of age/sex?

Results: The term age mates initially confused me a little, I thought it was a typo in the table. Perhaps "cohort" or similar accepted term?

Discussion: I feel you could just say "non-breeder" or "virgin" here.

Materials and methods:

I am also not quite sure about the use of simple paired t-tests to address a rather complex question, with many potential confounding variables. While I am all for using simple stats where possible, it seems like there would be more going on (age difference of a dyad, group size, age of orphan, year of study) that should be controlled for/investigated. Given the detailed models that follow, perhaps some clarity about why this is not necessary for the question currently being addressed would be useful?

Additionally, is there a citation for this permutation approach over permuting the networks themselves?

This suggests multiple models, but it seems only one model is described. I think I need to see these models/model formula laid out in a table for clarity. I'd also like to see a citation for the use of the random effects structure to control network independence.

Reviewer #2:

This is a novel, very well written and well-researched project, making use of a fantastic long-term dataset. It is interesting to a wide readership and of high scientific value. I do not really have any major concern but a few queries and suggestions in order improve the clarity in places.

List of slightly more substantial comments/queries:

1) Results – throughout – can you please include sample sizes when stats values are given?

2) Results – can you please also give us an idea of how much time they spent in proximity and contact? It’s good to see the stats but it would be nice to get a feel for who much time we are talking about (and how big the actual change is).

3) Discussion – I don't follow this argument – can you explain the reduced siring opportunities? Also, can you explain why infant-orphans are doing BETTER than older age orphans? This seems counter-intuitive and I am wondering why this would be the case?

4) Can you discuss a little who initiated the contact and proximity? Would that be on the initiative of the orphan or of the other group members?

5) Can you provide any thoughts on what happens after the 6 months? You are concluding that the strengthening of bonds for 6 months after the loss of maternal care is responsible for the lack of adverse effects – but do you have any evidence or indication that this strengthening is extending over a longer time period? I understand that you chose 6 months in order avoid other confounding variables in the analysis, such as changes to group composition – but in the discussion it reads like these bonds are strengthen “forever”. Some indicators that this indeed the case would be good to see if this generalization is warranted.

6) Materials and methods – a few more details about the data feeding into the networks would be good – which behaviours were included as affiliative behaviours? And how was the SRI calculated, ie did you use frequencies or durations of time in contact? More details on how focal sampling was carried out (length of focals and frequency of data recording) would be useful.

7) I would also like to see more details on number of groups, group sizes and compositions used in this study (maybe as supplementary material); all of this can go in supplementary material – but would be nice to have. Also, information on observation times for each social group and how stable they were would be helpful.

Reviewer #3:

The manuscript "Social groups buffer maternal loss in mountain gorillas" examines the potential consequences of maternal loss in mountain gorillas. The topic of early life adversity and consequences of maternal care that extends beyond weaning is of growing interest to researchers including behavioral ecologists and anthropologists. The results presented in the manuscript are an interesting and important contribution to that literature. Whereas most studies have found negative fitness consequences associated with maternal loss, the results reported here indicate that gorillas who experience early maternal loss do not face negative consequences in terms of survival, maturation (age at first birth), or an indicator of reproductive success (first offspring survival). Furthermore, rather than speculating the authors follow-up on social buffering as a potential explanation for this somewhat unexpected (given outcomes observed in other social species) result using social network analyses. My comments on the manuscript primarily concern organization and clarity to help the reader, particularly the reader of a broader journal such as *eLife*.

1) I realize that the structure of an *eLife* article has the methods at the end, but as currently presented most of the results are impossible to interpret without a lot of flipping back and forth searching for information. For example, the Results start out by briefly stating that survival was examined using a cox proportional hazards model, which was very helpful for interpreting the results. However, in the next section there is reference to model results without any information about what type of model it is. Later, there are t statistics and p values with no explanations and no indication they came from permutation-based tests.

2) Some of the abruptness of the transition from introduction to results might be helped by stating clearer predictions that help set up the outcomes you tested. I found the descriptions and expectations concerning the fitness outcomes clearer than the social network outcomes and suggests more information be given about the networks before the results of the analyses are presented.

3) I also have some questions concerning the social network analysis. One additional analysis that might be interesting is looking at binary degree along with weighted degree. High weighted degree can result from few strong connections or many weak connections and presenting both weighted and binary degree might indicate one strategy (find a strong buddy) versus another (cast a wide net). Furthermore, what was the variation in group size in these data? Did you take variation in group size into account when calculating network metrics. The maximum weighted degree of an individual in a group with 5 individuals is much lower than the maximum weighted degree of an individual in a group with 15 individuals.

4) Results: Regarding the results for dispersal – any reason to believe group size will influence likelihood of dispersal?

5) Materials and methods: Social buffering of maternal loss results: can you clarify whether the edge to mom included when weighted degree was calculated?

6) Table 5:

- How was age included in the GAMM models? Age in years?

- This table could be presented more clearly. Sometimes the bold is used to describe the two variables under it (Maternal orphan and Group member) and other times the bold itself is a variable with multiple levels under it. Maybe just write out "Orphan age" "Orphan Sex" and not include the extra bold rows?

- Should the age/sex class – sibling interaction results be relative to the dominant male who is or isn't a sibling?

- Where are the results of the smoothed term?

7) Figure 1: Curious about why each age category has its own line. Since maternal loss is a time varying covariate, shouldn't the survival probability of individuals who lost their mother at 7 (for example) be the same as non-orphans until age 7? Apologies if what I described above is the actually the case. It is tricky to tell where the dashed lines start.

8) Materials and methods: Can you explain why sampling variation warranted a smooth? Not necessarily questioning your decision – I find it interesting and looking for more information about it!

[Editors' note: further revisions were suggested prior to acceptance, as described below.]

Thank you for revising and resubmitting your article "Social groups buffer maternal loss in mountain gorillas" for consideration by *eLife*. Your article and response have been evaluated by a Guest Editor and Christian Rutz as the Senior Editor.

We would like to draw your attention to changes in our revision policy that we have made in response to COVID-19 (https://elifesciences.org/articles/57162). Specifically, we are asking Editors to quickly process manuscripts, like yours, that they judge can potentially stand as *eLife* papers without additional data. Thus, the revisions requested below are aimed at (a) addressing clarity and presentation of the manuscript, and (b) addressing further considerations about the current analyses and arising conclusions.

As outlined in the first decision letter, this manuscript provides an "in-depth assessment of the consequences of maternal loss for wild mountain gorillas" and was found to be a "rigorous and detailed examination" that was appreciated by the Editors and all the reviewers. During the first assessment by the reviewers and Editors, some potential problems with the wording of particular parts of the text were raised, and this revision appears to address all of these in full. However, the first assessment also raised some issues with the reported analyses, and unfortunately, there is still some lack-of-clarity and some issues that need to be resolved. To outline these issues in as clear a way as possible, the below text takes a 3-step approach for each issue: (A) it revisits the initial comment, (B) provides the authors' response (and associated manuscript text), and finally (C) describes why a problem still exists and what needs to be addressed.

1) Differences in social changes

1A) Revisiting the issue: "The use of paired t-tests for the examination of social changes is a bit confusing here. Specifically -on my part- I'm unsure how these properly control for repeated sampling of individuals when considering changes in dyadic association scores? Also, the changes reported here appear to just assess how orphan behaviour altered between the pre- and the post- maternal loss period, but how do we know that such changes aren't expected anyway over time? Ideally this analysis would directly compared orphaned to non-orphaned individuals. Note the comments from reviewer 1 about these paired t-tests need to be addressed too"

1B) Author response: "Yes, we entirely agree and have changed these analyses to instead compare whether network metrics (binary degree, weighted degree and eigenvector centrality) change differently in orphans relative to other immature gorillas within the same group during the same time period using node-based permutations constrained to swap only between individuals within the same incident of maternal loss (based on the same pair of networks). Described in the Materials and methods and Results."

1C) Current Issue:

While we appreciate that the authors have taken the first comment onboard and reconsidered the approach, there are two remaining problems with the current approach. Namely:

1Ci) In the Results, the results of these tests are reported. But, when reporting these results, it is important to not entirely focus on the p-value generated from the null models, but to also report the full results of the standard statistical test as well (i.e., the estimate, the SE, the t value, and the standard p-value). The p-value from the null model is simply used to 'double-check' the results of the standard statistical test, and shouldn't be fully relied upon for conclusions like this. For example, the first result reported here is "(t=1.721,p=0.029)" which isn't clear to the reader, as a t-value this low wouldn't generally result in a p-value this low. Instead, the text needs to report the actual observed statistical test results from the standard test (including the real p-value) and then also include a P_null_ value (the null value generated from the node permutations). If the primary standard statistical test shows no significant difference (as may be the case here), then this primary result should be relied upon for drawing conclusions (not the null model p-value, as this is mainly just for double-checking tests, rather than drawing direct conclusions from them in this particular way).

Also, as a small side-point, the method used here for actually calculating the node permutation p-value is quite a strange one. Specifically, it is stated in the revised text that "two-tailed p-values were calculated as two times the proportion of permutation t-values greater or smaller than the t-value from the real data set". But this is problematic and doesn't make clear sense? What is actually being done here?

1Cii) The text goes on to state that "within contact-based networks orphans did not show these same gains, with no significant increase in binary degree (t=0.099, p=0.730), weighted degree (t=1.110, p=0.397) or eigenvector centrality (t=0.129, p=0.741) relative to non-orphans". But it isn't clear where these results are actually derived from. We are assuming that they come from a subset of tests the authors refer to in their response, i.e., 'node-based permutations constrained to swap only between individuals within the same incident of maternal loss'. But, if the swaps are constrained between individuals who experience the same incident of maternal loss, and you subsequently try to look at the comparable difference between incident of maternal loss, then the tests will never find a difference because this is already controlled for earlier in the pipeline? Some clarity about what is actually going on here needs to be added to the manuscript (and also some thought about whether such tests are appropriate for testing the questions at hand).

2) Social networks and observations

2A) Revisiting the issue: The first assessment of the paper flagged up that animal social networks are strongly influenced by observation, and requested some analytical consideration of this in the revisions.

2B) Author Response: The authors responded by providing more details about the data collection methods, which is very much appreciated. However, the revision doesn't include any attempts to actually control for/integrate consideration of differences in observation number. Instead, the response states that the Simple Ratio Index includes this (stating "variation in the number of focal scans per individual which is accounted for within the simple ratio index" and "we did not believe that these extra data would bias the estimate of the specific relationships based on SRI (as these account for the total number of observations)".

2C) Current Issue: Unfortunately, the SRI does not control for differences in observation number in the way that the authors lay out here. While the SRI takes some of the difference in observation into account during the calculation of the dyadic values, it in no way fully accounts for this; it just uses it for the calculation. For instance, even in simulations where all individuals have the same social phenotype, those individuals which are observed more will have higher centrality (e.g., higher degree) than those which are observed less. This is simply a product of how these metrics are calculated and how these networks emerge. Therefore, the previous comments from the Editors and the reviewers that the analyses should directly consider 'observation number' still stands.

3) Outstanding analytical issues

3A/B) Revisiting the issue: In the reviewer comments, a few analytical (smaller) issues were pointed out, and in most cases the authors have addressed these fully and clearly. However, a couple of key points remain. Particularly, controlling for age and network differences within the models themselves, and also using 'change in metric' as the response.

3C) Current Issue: The revision didn't make any attempt to actually integrate age of individuals into the models, nor did it try to consider network groups, but the reviewers made solid points that these would be good to consider. We believe these should be integrated into the paper as supplementary analyses. Finally, the switch to considering “change in metric” as the response is certainly useful in lots of ways, but have the authors' considered problems with “regression to the mean” in this sense? Specifically, we can automatically expect this metric to be altered significantly by the initial value, whereby those which initially have a large SN metric value are likely to (just by chance) have large -ve change values, and whereby those which initially have a small SN metric value are likely to (just by chance) have large +ve change values. Have the authors assessed this as a driver/contributor to the results here? (Note this consideration relates to the above point about considering things like age, and network group, which might alter these initial values and thus affect the change values.)

---

## [Author Response]

Essential revisions:We judge that the manuscript would benefit from addressing the reviewers' comments point-by-point, and have therefore decided to append their full reports below. In relation to some of the comments detailed below, we would like to highlight the following three points:1) The use of paired t-tests for the examination of social changes is a bit confusing. Specifically, we are unsure how these properly control for repeated sampling of individuals when considering changes in dyadic association scores. Also, the changes reported here appear to just assess how orphan behaviour altered between the pre- and the post-maternal loss period, but how do we know that such changes aren't expected anyway over time? Ideally, these analyses would directly compare orphaned to non-orphaned individuals. Note the comments from reviewer #1 about these paired t-tests that also need to be addressed.

Yes, we entirely agree and have changed these analyses to instead compare whether network metrics (binary degree, weighted degree and eigenvector centrality) change differently in orphans relative to other immature gorillas within the same group during the same time period using node-based permutations constrained to swap only between individuals within the same incident of maternal loss (based on the same pair of networks). Described in the Materials and methods and Results.

2) Animal social networks (either dyadic association scores or individual-level metrics) are strongly influenced by observation. But it is not clear how these analyses controlled for, or considered, how individual-level differences in observation propensity might alter inferences of their social network position/association scores, and what this might mean for the arising conclusions. We think the manuscript would benefit from including a much more detailed report of how individuals differed in their likelihood of being observed in relation to the social behaviour considered here, and how this may shape the arising networks, and whether controlling for these effects might alter the results/conclusions.

We have added further details of the sampling methods to the manuscript. Data were collected on fully habituated mountain gorilla groups using focal sampling. Under this approach researchers systematically worked their way through a randomly ordered list of all individuals in the group. If an individual could not be observed, the researchers moved on to the next individual on the list and returned to them the subsequent day. During focals, researchers recorded all individuals within 2m and within physical contact of the focal individual every 10 minutes for an hour. This meant there was no bias in certain individuals being missed that were within 2m or in physical contact of the focal individual as with these small distances and extended periods of observation it is extremely unlikely any individuals at this close proximity were missed. We therefore do not believe there are any biases in the likelihood of certain individuals in a group being observed when participating in the social behaviour considered here, except the variation in the number of focal scans per individual which is accounted for within the simple ratio index.

During the 17 years of social data used in these analyses, some additional projects took place. For example, one project examining the relationships of dominant males and infants. This meant that additional focal data were available for certain individuals and led to some variation in the number of focals per individual within a group. However, we did not believe that these extra data would bias the estimate of the specific relationships based on SRI (as these account for the total number of observations) but would likely just lead to more accurate estimates of these relationships relative to others in the group. We therefore kept these extra focal data within the larger dataset but included the number of focals of either individual involved in a pairwise relationship as a smoothing term in the GAMMs in case there was some potential non-linear bias introduced by this sample size variation that we hadn’t thought of.

3) We are unsure why only weighted degree was used as a network metric; reviewer #3 also raises a point about this metric. We suggest that this could be addressed through also considering mean average non-zero dyadic bond scores as a metric. Furthermore, you may want to re-run the network metric-based analyses, but with other measures of social behaviour (e.g., eigenvector centrality) to get an idea of how wider indirect centrality within the network may relate to maternal loss.

Thanks so much for these suggestions, we think they add really useful information on the overall network changes faced by orphans. We have rerun the analyses looking at the change in binary degree, weighted degree and eigenvector centrality. We used mixed models to account for differences between networks and permuted only between individuals within the same network (as discussed for point 1). There was no relationship between change in weighted degree and network size so we therefore stuck with weighted degree as the more commonly used metric rather than changing to average non-zero dyadic bond scores.

Reviewer #1:This is an interesting paper about how the loss of a mother influences individuals future development, or rather the fact that it might not due to social connections with other group members. The paper examines a very comprehensive set of question, and on the whole I found it very well written, though I have some quibbles about how certain ideas are presented/supported. The dataset presented seems impressive. The methods used generally seem appropriate and robust, though I'd like to see some extra justifications for some of the approaches taken during network analysis. For more info, see detailed comments:IntroductionI think an example of how the presence of the mother affects fitness in later life would be nice here, in order to lend weight to the statement about maternal loss having affects throughout an individual's lifetime. The third paragraph contains examples, and currently feels repetitive in terms of the points raised.

Thank for this suggestion – we fully agree and have reorganised the Introduction considerably in an attempt to make it more streamlined and less repetitive

Similarly, I think "these negative changes" could be expanded on in the preceding sentences with an example of negative changes to the social environment.

This has now been addressed in the reorganization of the Introduction.

I am not sure how convinced I am that the human example here adds to the authors' point. As mentioned above, I would prefer to see more general examples of these ideas/effects in social mammals (or animals in general). If the authors' really want to keep a human example, I'd like to see significantly more detail about the circumstances in which this result was obtained.

This has been removed – we agree that it does not tie in very well here.

It feels slightly odd to discuss sex specific consequences before the general consequences.

Agreed – Introduction reorganized as discussed above.

As above, I am slightly unconvinced by the need/relevance to relate these results back to humans. This might be a result of me not having an anthropology background of course. Related to this, I feel this is making a rather large assumption about the familiarity a reader might have with this literature. Some small details about what these populations are might help convince that they are relevant to build the argument in the same paragraph as killer whales. There are a few other instances of this in the Discussion too, but I'll avoid repeating myself.

Thank you for pointing this issue out. We have now added further information to the Introduction to explain why humans are an important example – they are potentially the only social mammal where social buffering does overcome the social adversity of maternal loss.

Any directional predictions going into the study? There are quite a few variables under investigation, and it would be nice to link them more strongly to the ideas articulated in the Introduction. Is social buffering more likely in certain group compositions of age/sex?

We have now more clearly stated our hypotheses going into the study to the Introduction. We hypothesize that as demonstrated in chimpanzees, gorillas may face greater fitness costs if they suffer maternal loss at an earlier age and that males may face greater costs from maternal loss than females due to their longer periods of mother-offspring co-residence. Alternatively, as observed in many human populations, the cohesive, stable social groups of mountain gorillas may enable social buffering from group members to compensate for the social costs of maternal loss with minimal fitness consequences to maternal loss. In particular, dominant males may take on crucial roles in buffering the social adversity faced by maternal orphans, as past research has demonstrated the strong bond between dominant males and young orphans who may regularly share a nest at night.

Results: The term age mates initially confused me a little, I thought it was a typo in the table. Perhaps "cohort" or similar accepted term?

This term was used as it is the same term used in the elephant paper investigating social responses to maternal loss. It’s also quite a common term in the human literature although is slightly confusing as it has nothing to do with mating. We’ve therefore added in “those within 2 years age of the orphan” where it is first used.

Discussion: I feel you could just say "non-breeder" or "virgin" here.

We have changed it to “pre-reproductive”. We thought that non-breeder could be confused with infertile older females and virgin had too many human social connotations.

Materials and methodsI am also not quite sure about the use of simple paired t-tests to address a rather complex question, with many potential confounding variables. While I am all for using simple stats where possible, it seems like there would be more going on (age difference of a dyad, group size, age of orphan, year of study) that should be controlled for/investigated. Given the detailed models that follow, perhaps some clarity about why this is not necessary for the question currently being addressed would be useful?Additionally, is there a citation for this permutation approach over permuting the networks themselves?

Yes, we agree there are some issues with this previous t-test approach. We have now altered this analysis to compare the change in network metric between the pre- and post-maternal loss periods, testing whether orphans and non-orphans show different changes in network metrics (as described above) using node-based permutations between immature gorillas (both orphans and non-orphans) within the same group (Described in the Materials and methods and Results.). This tests the observed pattern against a null model in which the network metrics of orphans and non-orphans within the same group and the same set of networks do not change differently after the maternal loss incident. This therefore controls for differences in group size, group composition, age-related changes over time, year of study etc. This does not control for the age of the immature gorilla themselves, however ages were distributed fairly evenly across both orphans and non-orphans (the mean age ± SD of orphans was 5.12 ±1.49 years and for non-orphans was 4.71 ±1.71 years) and there is therefore no reason to believe that age could be driving a difference between orphans and non-orphans.

This suggests multiple models, but it seems only one model is described. I think I need to see these models/model formula laid out in a table for clarity. I'd also like to see a citation for the use of the random effects structure to control network independence.

Model formulas have now been added to the table legends in which results are reported (Table 5 and Supplementary file 4 and Supplementary file 5).

The random effects structures are used to control for the non-independence of pairwise relationships involving the same individual but each pairwise relationship is only included in each model once. Apologies, this section was poorly worded and was been altered (subsection “Relationship changes following maternal loss”). We don’t believe that random effect structures would adequately account for non-independence of network metrics, hence the permutation approach used for degree and centrality metrics.

Reviewer #2:This is a novel, very well written and well-researched project, making use of a fantastic long-term dataset. It is interesting to a wide readership and of high scientific value. I do not really have any major concern but a few queries and suggestions in order improve the clarity in places.List of slightly more substantial comments/queries:1) Results – throughout – can you please include sample sizes when stats values are given?

Apologies, this has now been added more consistently throughout.

2) Results – can you please also give us an idea of how much time they spent in proximity and contact? It’s good to see the stats but it would be nice to get a feel for who much time we are talking about (and how big the actual change is).

We have added in this information for mother-offspring dyads to give a feel for exactly what losing that bond means for their time in proximity and their affiliative contact. Changes overall for the group are harder to interpret as it’s the sum of the proportion of time spent in contact or proximity of another individual so a value of 1 could mean they are always in physical contact with only 1 other individual or that they are in physical contact with 2 other individuals simultaneously but only half the time. We have therefore included this information as a percentage increase or decrease compared to the pre-maternal loss period (subsection “Changes in network position following maternal loss”).

3) Discussion – I don't follow this argument – can you explain the reduced siring opportunities?

We have reworded this section to clarify – other studies have demonstrated that for males, those that remain in their natal group have greater siring opportunities than those that disperse. If they remain, they can potentially mate while they are subordinate (although far less so than the dominant male) and they can also wait it out for the dominant male to die or become weak so they can eventually take over. In contrast, males that disperse will not be accepted into another group and are solitary until they are able to attract females and form a group of their own. Many solitary males never succeed in attracting females and therefore have almost no siring opportunities.

Also, can you explain why infant-orphans are doing BETTER than older age orphans? This seems counter-intuitive and I am wondering why this would be the case?

Further details have been added (Discussion). We believe it may be due to two things. Firstly, that individuals that suffer maternal loss at a younger age show greater strengthening of proximity-based relationships with group members. Secondly, that there is a greater period of time between maternal loss and the age at which they could disperse. Both of these things mean that these infant-orphaned gorillas may have stronger social relationships within their group by the time they reach the age at which they may choose to disperse.

4) Can you discuss a little who initiated the contact and proximity? Would that be on the initiative of the orphan or of the other group members?

We cannot say from this data who initiated the contact or proximity, which is an important caveat to the study. This has now been discussed further (Materials and methods).

5) Can you provide any thoughts on what happens after the 6 months? You are concluding that the strengthening of bonds for 6 months after the loss of maternal care is responsible for the lack of adverse effects – but do you have any evidence or indication that this strengthening is extending over a longer time period? I understand that you chose 6 months in order avoid other confounding variables in the analysis, such as changes to group composition – but in the discussion it reads like these bonds are strengthen “forever”. Some indicators that this indeed the case would be good to see if this generalization is warranted.

Thank you for highlighting this important caveat, we have now added some discussion of this into the manuscript (Materials and methods).

6) Materials and methods – a few more details about the data feeding into the networks would be good – which behaviours were included as affiliative behaviours? And how was the SRI calculated, ie did you use frequencies or durations of time in contact? More details on how focal sampling was carried out (length of focals and frequency of data recording) would be useful.

More details on the behaviours included as affiliative contact have now been added to the Results section where they are first mentioned. Further details on the focal sampling approach and the use of frequencies rather than durations of association have now been added to the Materials and methods.

7) I would also like to see more details on number of groups, group sizes and compositions used in this study (maybe as supplementary material); all of this can go in supplementary material – but would be nice to have. Also, information on observation times for each social group and how stable they were would be helpful.

This information has been added to the supplementary material (Supplementary file 4).

Reviewer #3:The manuscript "Social groups buffer maternal loss in mountain gorillas" examines the potential consequences of maternal loss in mountain gorillas. The topic of early life adversity and consequences of maternal care that extends beyond weaning is of growing interest to researchers including behavioral ecologists and anthropologists. The results presented in the manuscript are an interesting and important contribution to that literature. Whereas most studies have found negative fitness consequences associated with maternal loss, the results reported here indicate that gorillas who experience early maternal loss do not face negative consequences in terms of survival, maturation (age at first birth), or an indicator of reproductive success (first offspring survival). Furthermore, rather than speculating the authors follow-up on social buffering as a potential explanation for this somewhat unexpected (given outcomes observed in other social species) result using social network analyses. My comments on the manuscript primarily concern organization and clarity to help the reader, particularly the reader of a broader journal such as eLife.1) I realize that the structure of an eLife article has the methods at the end, but as currently presented most of the results are impossible to interpret without a lot of flipping back and forth searching for information. For example, the Results start out by briefly stating that survival was examined using a cox proportional hazards model, which was very helpful for interpreting the results. However, in the next section there is reference to model results without any information about what type of model it is. Later, there are t statistics and p values with no explanations and no indication they came from permutation-based tests.

Thank you for pointing this out – this additional information has now been added throughout the Results section.

2) Some of the abruptness of the transition from introduction to results might be helped by stating clearer predictions that help set up the outcomes you tested. I found the descriptions and expectations concerning the fitness outcomes clearer than the social network outcomes and suggests more information be given about the networks before the results of the analyses are presented.

Agreed – thanks for this suggestion, we have now added this information at the end of the Introduction and prior to the network results (Results).

3) I also have some questions concerning the social network analysis. One additional analysis that might be interesting is looking at binary degree along with weighted degree. High weighted degree can result from few strong connections or many weak connections and presenting both weighted and binary degree might indicate one strategy (find a strong buddy) versus another (cast a wide net). Furthermore, what was the variation in group size in these data? Did you take variation in group size into account when calculating network metrics. The maximum weighted degree of an individual in a group with 5 individuals is much lower than the maximum weighted degree of an individual in a group with 15 individuals.

Thank you very much for these constructive suggestions. We have reanalyzed this section entirely incorporating binary degree and eigenvector centrality (Described in the Materials and methods and Results). By looking at the change in these metrics rather than the raw values we minimize the issues relating to variation in group size, but also control for these by including random effects for each network and only permuting between individuals within the same network to generate p-values. Data on group sizes are provided in Supplementary file 4 and the Figure 2—source data 1.

4) Results: Regarding the results for dispersal – any reason to believe group size will influence likelihood of dispersal?

No evidence of this was found in previous studies so we have not included it in the models in the interests of simplicity. Robbins, (1995). Stoinski et al., (2009).

5) Materials and methods: Social buffering of maternal loss results: can you clarify whether the edge to mom included when weighted degree was calculated?

This has now been reanalyzed, as described above. The edge to the mother was not included when looking at the overall change in network metrics and clarification of this has been added. Only broad descriptive statistics are used for the mother-offspring bond of orphans to give an idea of the importance of that relationship pre-maternal loss.

6) Table 5:- How was age included in the GAMM models? Age in years?

Yes – added.

- This table could be presented more clearly. Sometimes the bold is used to describe the two variables under it (Maternal orphan and Group member) and other times the bold itself is a variable with multiple levels under it. Maybe just write out "Orphan age" "Orphan Sex" and not include the extra bold rows?

This has been reformatted.

- Should the age/sex class – sibling interaction results be relative to the dominant male who is or isn't a sibling?

Yes, this has now been added.

- Where are the results of the smoothed term?

These have now been added below the table.

7) Figure 1: Curious about why each age category has its own line. Since maternal loss is a time varying covariate shouldn't the survival probability of individuals who lost their mother at 7 (for example) be the same as non-orphans until age 7? Apologies if what I described above is the actually the case. It is tricky to tell where the dashed lines start.

You are correct that we used time varying covariates for the Cox-proportional hazards model, however we still distinguished between the three categories to make our study comparable to other studies (e.g., Stanton et al., 2020). The analysis shows two things, that there are no proportional differences between the three orphan classes and the non-orphans, therefore a proportional hazards is not the correct assumption, and second, when running the Bayesian model, we confirmed that there are in fact no differences between the orphan classes.

To make the points at which orphans and non-orphans split clearer we have added Figure 1—figure supplement 1 which plots each age category by sex separately and indicates the point from which their survival is modelled separately.

8) Materials and methods: Can you explain why sampling variation warranted a smooth? Not necessarily questioning your decision – I find it interesting and looking for more information about it!

We think it is probably unnecessary but accounts for any potential non-linear relationship between sampling and relationship strength. The SRI index takes into account any linear relationship, but we wanted to be extra cautious because of some of the uneven sampling introduced by different focal sampling protocols over the years (as described in more detail above).

[Editors' note: further revisions were suggested prior to acceptance, as described below.]

As outlined in the first decision letter, this manuscript provides an "in-depth assessment of the consequences of maternal loss for wild mountain gorillas" and was found to be a "rigorous and detailed examination" that was appreciated by the Editors and all the reviewers. During the first assessment by the reviewers and Editors, some potential problems with the wording of particular parts of the text were raised, and this revision appears to address all of these in full. However, the first assessment also raised some issues with the reported analyses, and unfortunately, there is still some lack-of-clarity and some issues that need to be resolved. To outline these issues in as clear a way as possible, the below text takes a 3-step approach for each issue: (A) it revisits the initial comment, (B) provides the authors' response (and associated manuscript text), and finally (C) describes why a problem still exists and what needs to be addressed.1) Differences in social changes1A) Revisiting the issue: "The use of paired t-tests for the examination of social changes is a bit confusing here. Specifically -on my part- I'm unsure how these properly control for repeated sampling of individuals when considering changes in dyadic association scores? Also, the changes reported here appear to just assess how orphan behaviour altered between the pre- and the post- maternal loss period, but how do we know that such changes aren't expected anyway over time? Ideally this analysis would directly compared orphaned to non-orphaned individuals. Note the comments from reviewer 1 about these paired t-tests need to be addressed too"1B) Author response: "Yes, we entirely agree and have changed these analyses to instead compare whether network metrics (binary degree, weighted degree and eigenvector centrality) change differently in orphans relative to other immature gorillas within the same group during the same time period using node-based permutations constrained to swap only between individuals within the same incident of maternal loss (based on the same pair of networks). Described in Materials and methods and Results."1C) Current Issue:While we appreciate that the authors have taken the first comment onboard and reconsidered the approach, there are two remaining problems with the current approach. Namely:1Ci) In the Results, the results of these tests are reported. But, when reporting these results, it is important to not entirely focus on the p-value generated from the null models, but to also report the full results of the standard statistical test as well (i.e., the estimate, the SE, the t value, and the standard p-value). The p-value from the null model is simply used to 'double-check' the results of the standard statistical test and shouldn't be fully relied upon for conclusions like this. For example, the first result reported here is "(t=1.721,p=0.029)" which isn't clear to the reader, as a t-value this low wouldn't generally result in a p-value this low. Instead, the text needs to report the actual observed statistical test results from the standard test (including the real p-value) and then also include a P_null_ value (the null value generated from the node permutations). If the primary standard statistical test shows no significant difference (as may be the case here), then this primary result should be relied upon for drawing conclusions (not the null model p-value, as this is mainly just for double-checking tests, rather than drawing direct conclusions from them in this particular way).

Thank you for clarifying this point. We have now included the full model outputs as supplementary tables and reported the estimate, SE, t-value, standard p-value and P_null_, and adjusted the wording to focus less on the P_null_. (Results). Standard p-value and P_null_ values correspond closely overall. With changes in the overall model (described below in our response to sections 2 and 3) results remain very similar. Metrics from networks based on affiliative contact do not increase in orphans relative to non-orphans. The eigenvector centrality of orphans in networks based on proximity still shows a strong increase relative to non-orphans (Est=0.169±0.037, t=4.594, P<0.001, P_null_<0.001), whilst weighted degree also increases but to a lesser extent (Est=0.075±0.032, t=2.337, P=0.021, P_null_=0.024). Binary degree of orphans no longer increases significantly relative to non-orphans in these proximity-based networks.

Also, as a small side-point, the method used here for actually calculating the node permutation p-value is quite a strange one. Specifically, it is stated in the revised text that "two-tailed p-values were calculated as two times the proportion of permutation t-values greater or smaller than the t-value from the real data set". But this is problematic and doesn't make clear sense? What is actually being done here?

We have reworded this section to make the approach we have used clearer. We used a two-tailed approach since we were interested in both whether the observed test statistic was larger than those in the null models (ie strengthening of relationships between orphans and other group members after maternal loss lead to increased network metrics) or smaller than those in the null models (ie orphans relationships with other group members weakened after losing their mothers relative to other immature gorillas in the group leading to decreased network metrics). If the observed t-value was below the median of the null model t-values, P_null_ was calculated as:

2 x numberofnulltvalueslowerthanobservedt−valuetotal null t values

If the observed t-value was above the median of the null model t-values, P_null_ was calculated as:

2 x numberofnulltvaluesgreaterthanobservedt−valuetotal null t values

Such that a p-value of <0.05 would indicate the observed t-value was within the lowest 2.5% of null values or highest 2.5% of null values.

1Cii) The text goes on to state that "within contact-based networks orphans did not show these same gains, with no significant increase in binary degree (t=0.099, p=0.730), weighted degree (t=1.110, p=0.397) or eigenvector centrality (t=0.129, p=0.741) relative to non-orphans". But it isn't clear where these results are actually derived from. We are assuming that they come from a subset of tests the authors refer to in their response, i.e., 'node-based permutations constrained to swap only between individuals within the same incident of maternal loss'. But, if the swaps are constrained between individuals who experience the same incident of maternal loss, and you subsequently try to look at the comparable difference between incident of maternal loss, then the tests will never find a difference because this is already controlled for earlier in the pipeline? Some clarity about what is actually going on here needs to be added to the manuscript (and also some thought about whether such tests are appropriate for testing the questions at hand).

Apologies –this was not worded very clearly by us. When we refer to “the same incident of maternal loss” we are meaning individuals within the same group when a group-member suffered maternal loss (same group, same time period, same set of networks) rather than specifically only swapping between individuals that suffered maternal loss themselves. Ie if there were 5 immature gorillas within a group and one suffered maternal loss, our node-based permutations would permute the orphan/non-orphan labels associated with each of those immature gorillas in that network. We have tried to make this clearer in the manuscript. These values come from the same models discussed above which are run twice: once on networks based on 2m proximity and once on networks based on affiliative contact. This has now been clarified further in the manuscript.

2) Social networks and observations2A) Revisiting the issue: The first assessment of the paper flagged up that animal social networks are strongly influenced by observation, and requested some analytical consideration of this in the revisions.2B) Author Response: The authors responded by providing more details about the data collection methods, which is very much appreciated. However, the revision doesn't include any attempts to actually control for/integrate consideration of differences in observation number. Instead, the response states that the Simple Ratio Index includes this (stating "variation in the number of focal scans per individual which is accounted for within the simple ratio index" and "we did not believe that these extra data would bias the estimate of the specific relationships based on SRI (as these account for the total number of observations)".2C) Current Issue: Unfortunately, the SRI does not control for differences in observation number in the way that the authors lay out here. While the SRI takes some of the difference in observation into account during the calculation of the dyadic values, it in no way fully accounts for this; it just uses it for the calculation. For instance, even in simulations where all individuals have the same social phenotype, those individuals which are observed more will have higher centrality (e.g., higher degree) than those which are observed less. This is simply a product of how these metrics are calculated and how these networks emerge. Therefore, the previous comments from the Editors and the reviewers that the analyses should directly consider 'observation number' still stands.

Thank you for pointing this out. All network metric models now include a smoothing term for the number of focal scans per individual to account for possible increases in network metric with number of observations of an individual (described in the Materials and methods). A smoothing term was chosen as these relationships may not be linear, particularly at higher numbers of observations. The models of pairwise relationships include the total number of focals of both individuals involved in the relationship as a smoothing term (as was the case in the previous draft).

3) Outstanding analytical issues3A/B) Revisiting the issue: In the reviewer comments, a few analytical (smaller) issues were pointed out, and in most cases the authors have addressed these fully and clearly. However, a couple of key points remain. Particularly, controlling for age and network differences within the models themselves, and also using 'change in metric' as the response.3C) Current Issue: The revision didn't make any attempt to actually integrate age of individuals into the models, nor did it try to consider network groups, but the reviewers made solid points that these would be good to consider. We believe these should be integrated into the paper as supplementary analyses. Finally, the switch to considering 'change in metric' as the response is certainly useful in lots of ways, but have the authors' considered problems with 'regression to the mean' in this sense? Specifically, we can automatically expect this metric to be altered significantly by the initial value, whereby those which initially have a large SN metric value are likely to (just by chance) have large -ve change values, and whereby those which initially have a small SN metric value are likely to (just by chance) have large +ve change values. Have the authors assessed this as a driver/contributor to the results here? (Note this consideration relates to the above point about considering things like age, and network group, which might alter these initial values and thus affect the change values.)

Thank you for these suggestions. We have rerun all models of network metric change before and after maternal loss incorporating these suggestions. Age has now been directly included in the models and the metrics themselves have been adjusted relative to the specific network so that binary and weighted degree of each individual is proportional to the maximum value of binary or weighted degree within the network (as is already the case for eigenvector centrality), described in the Materials and methods. This should make network metrics more comparable across networks, but we also still included the network (or incident of maternal loss as referred to in earlier drafts) as a random factor in all models. The potential for initial values that are far from the mean to be driving the results is also now accounted for by including the deviance of the initial network metric value from the group mean (separate mean values calculated for each network as this was included as a random factor).

New Model run as a GAMM to allow inclusion of a smoothing term for number of focal scans:

Metric Change ~ Orphan + Age + Deviance of initial value from group mean + s(focal scans), random=~(1|Network))

As described in our response to section 1, these model changes did not alter the overall findings except that the increase in binary degree of orphans in proximity-based networks went from being weakly significant to no longer significant. This however still supports our previous conclusions that orphans primarily strengthen existing social bonds following maternal loss rather than forming new ones.